# Geometric Transformer with Interatomic Positional Encoding

**Yusong Wang**[1,2*†], **Shaoning Li**[2,3,4*†], **Tong Wang**[2‡], **Bin Shao**[2]
**Nanning Zheng**[1], **Tie-Yan Liu**[2]

[1] National Key Laboratory of Human-Machine Hybrid Augmented Intelligence,
National Engineering Research Center for Visual Information and Applications,
and Institute of Artificial Intelligence and Robotics, Xi'an Jiaotong University
[2] Microsoft Research AI4Science
[3] Mila - Québec AI Institute [4] Université de Montréal
wangyusong2000@stu.xjtu.edu.cn, nnzheng@mail.xjtu.edu.cn
shaoning.li@umontreal.ca
{watong, binshao, tyliu}@microsoft.com

## Abstract

The widespread adoption of Transformer architectures in various data modalities has opened new avenues for the applications in molecular modeling. Nevertheless, it remains elusive that whether the Transformer-based architecture can do molecular modeling as good as equivariant GNNs. In this paper, by designing Interatomic Positional Encoding (**IPE**) that parameterizes atomic environments as Transformer's positional encodings, we propose **Geoformer**, a novel geometric Transformer to effectively model molecular structures for various molecular property prediction. We evaluate Geoformer on several benchmarks, including the QM9 dataset and the recently proposed Molecule3D dataset. Compared with both Transformers and equivariant GNN models, Geoformer outperforms the state-of-the-art (SoTA) algorithms on QM9, and achieves the best performance on Molecule3D for both random and scaffold splits. By introducing IPE, Geoformer paves the way for molecular geometric modeling based on Transformer architecture. Codes are available at https://github.com/microsoft/AI2BMD/tree/Geoformer.

## 1 Introduction

Transformer [46] has been a dominant architecture in modeling various data modalities such as natural language, images, and videos. As such, it is natural to generalize Transformers in molecule modeling. Molecules are represented as either 2D topology structures or 3D geometric structures. Prevailing algorithms for modeling 2D topology [52, 36, 20] employ Transformers with global attention, which treat the molecular structures as fully connected graphs and devise a diverse range of positional encoding schemes. In contrast to topological graphs, geometric structures offer more comprehensive description of molecules by providing the information of 3D coordinates. There have been some recent attempts [28, 36, 55, 19, 29, 30, 50, 22, 8], to integrate specific geometric features into Transformers. For example, Transformer-M [28] considers pairwise distances as a learnable bias added to the attention weights as a supplement to the 2D topology encoding used in Graphormer [52]. Nevertheless, they are insufficient to comprehensively encompass the intricacies of the entire 3D molecular space, as they solely rely on distance information for positional encoding. In

---

[*]Work done during an internship at Microsoft Research.

[†]Equal contribution.

[‡]Corresponding author.

37th Conference on Neural Information Processing Systems (NeurIPS 2023).

contrast, modern GNNs emphasize the significance of *equivariance* as a crucial bias, and explore various methods to encode geometric information, including distances, angles, and dihedral angles [13, 12, 39, 26, 49, 47]. Several works [3, 31, 2] further utilize high-order geometric tensors to yield internal features in models, ensuring equivariance with respect to the E(3)/SE(3) group. As a result, the performance of EGNNs in molecular property prediction significantly surpasses that of methods based on Transformers.

The success of EGNNs underscores the advantage of integrating directional geometric information into neural networks for molecular modeling [12, 26, 47, 48], while employing such information in the Transformer-based architecture has yet to be developed. Intuitively, all geometric information is embedded in atomic coordinates, which can be naturally utilized as a bias for developing an effective positional encoding in Transformers. Based on atomic coordinates, atomic cluster expansion (ACE) theory [10, 21] is a complete descriptor to represent the environment of centered atoms . In this study, we first design interatomic positional encoding (IPE) by introducing cluster merging based on ACE theory. By incorporating IPE into traditional Transformers, we extend the capabilities of the Transformer, in terms of **Geoformer**, to effectively model molecular structures for molecular property prediction. We conduct a comprehensive evaluation of Geoformer using several benchmarks, which includes QM9 dataset [35] comprising 12 molecular properties and the recently proposed Molecule3D dataset [51], containing 3,899,647 molecules sourced from PubChemQC[32]. Our results demonstrate that Geoformer surpasses state-of-the-art algorithms on the majority of properties on QM9 dataset, and achieves the lowest mean absolute errors on Molecule3D for both random and scaffold splits. We also provide visualizations of the learned IPE, which shows that IPE can capture different positional information compared with PE that only encodes pairwise distances.

Our contributions can be summarized as follows:

- We introduce a novel positional encoding method, i.e., Interatomic Positional Encoding (IPE) to parameterize atomic environments in Transformer.

- By incorporating IPE, we propose a Geometric Transformer, in terms of **Geoformer**, which models valuable geometric information beyond pairwise distances for Transformer-based architecture.

- The proposed **Geoformer** achieves superior performance on molecular property predictions compared with Transformers and EGNNs.

## 2   Preliminary

The Atomic Cluster Expansion (ACE) [10] is a complete descriptor of the local atomic chemical environments, represented by hierarchical many-body expansion. The key components of ACE are: a) ACE defines a *complete* set of basis functions for the environment of the centered atom (radial basis functions and spherical harmonics in practice); b) ACE significantly reduces the computational efforts for body order computing to *linear* time complexity scaling with the number of atoms within molecule. These advantages serve ACE as an accurate, fast, and transferable theory framework for molecular modeling. In order to facilitate readers' comprehension of our interatomic positional encoding, we would present the core operations of ACE in several equations.

ACE includes a set of orthogonal basis functions $\phi_v(\hat{r}_{ij})$ to describe the spatial relations between two atoms. $\hat{r}_{ij}$ denotes the relative position pointing from atom $i$ to atom $j$, and $v$ indicates the functions' polynomial degree. ACE focuses on modeling the potential energy of a system by focusing on a collection of atoms, specifically atomic clusters. In this method, the centered atom, denoted as $i$, is surrounded by $K$ neighboring atoms, which form the atomic cluster. The potential energy of the atomic cluster depends on the hierarchical interactions between the central atom $i$ and its $K$ neighboring atoms, known as many-body expansion. The expansion of atomic potential energy could be written by:

$$E_i = \sum_{j_1}\sum_{v_1} c_{v_1}\phi_{v_1}(\hat{r}_{ij_1}) + \sum_{j_1 j_2}\sum_{v_1 v_2} c_{v_1 v_2}\phi_{v_1}(\hat{r}_{ij_1})\phi_{v_2}(\hat{r}_{ij_2}) + \cdots \tag{1}$$

with an unrestricted summation. $c_v$ indicates the expansion coefficients. However, the sum of surrounding neighbors within the cluster would scale to $\mathcal{O}(N^K)$ with $K$ neighbors and become

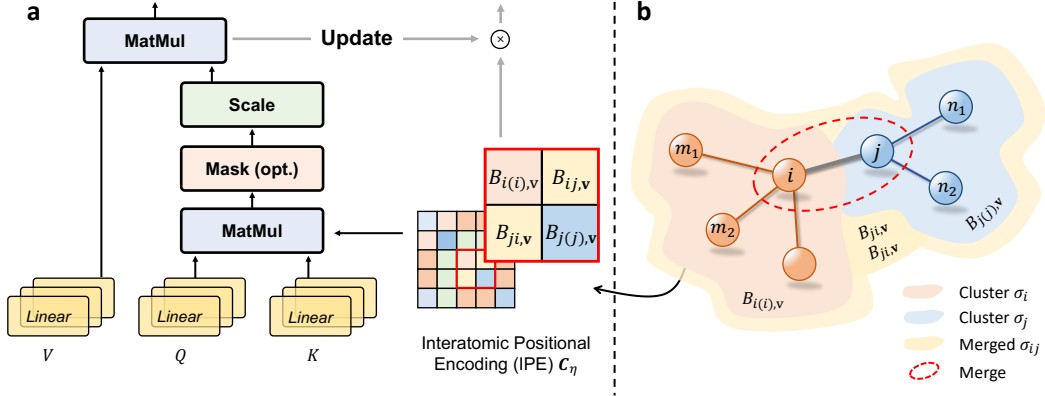

Figure 1: **Illustration of Interatomic Positional Encoding (IPE) $C_\eta$.** Panel **a** depicts the extended Self-Attention with $C_\eta$. $C_\eta$ contributes to the construction of attention weights while simultaneously being updated by atomic features; Panel **b** highlights the relationship between $C_\eta$, cluster $\sigma_i$, cluster $\sigma_j$, and the merged cluster $\sigma_{ij}$, as described in Theorem 1 and 2, Equation 6 to Equation 11.

numerically expensive. $N$ denotes the number of atoms within one molecule. ACE leverages the *density trick* to reduce the computational overhead. It defines *atomic base* $A_{i,v}$ and *A-basis* $A_{i,\mathbf{v}}$ as:

$$A_{i,v} = \sum_{j \in N(i)} \phi_v(\hat{r}_{ij}) \tag{2}$$

$$A_{i,\mathbf{v}} = \prod_{t=1}^{\epsilon} A_{i,v_t}, \quad \mathbf{v} = (v_1, \ldots, v_\epsilon) \tag{3}$$

where $\epsilon$ denotes the order of body expansion, i.e., $(\epsilon + 1)$-body expansion, and $\mathbf{v}$ stands for the set of $v$. By firstly summing the neighboring basis functions and then applying multiplication, ACE can efficiently represent the atomic expansion scaling with the complexity $\mathcal{O}(N)$. Since $A_{i,\mathbf{v}}$ is not rotationally invariant, we need additional Clebsch-Gordan coefficients $C_v$ to construct fully permutation and isometry-invariant basis functions (*B-basis*), and describe the potential of cluster $\sigma_i$ in Equation 1 as their linear combination with $c$-coefficients [21]:

$$B_{i,\mathbf{v}} = \sum_{\mathbf{v}'} C_{\mathbf{v}\mathbf{v}'} A_{i,\mathbf{v}'} \tag{4}$$

$$E_i = \sum_{\epsilon} c_{i,\mathbf{v}} B_{i,\mathbf{v}} = \mathbf{c}_i \cdot \mathbf{B}_i \tag{5}$$

## 3 Methods

In this section, we would introduce our Interatomic Positional Encoding (IPE) based on ACE theory in Section 3.1, and discuss how we integrate IPE into the Transformer architecture in Section 3.2.

### 3.1 Positional Encoding for Geometric Molecule Modeling

In the context of geometric molecule modeling, the positions of atoms are the most intuitive positional information to encode in Transformers (termed as "positional encoding"). While most recent works have adopted pairwise distances between atoms as the relative PE, such a representation is often inadequate for capturing the complex interactions within molecules. As a result, it is essential to use a more comprehensive and appropriate PE for geometric data. Motivated by ACE, we propose an Interatomic Positional Encoding (**IPE**) to efficiently describe the many-body contributions in Transformers for geometric molecule modeling. The distinction from ACE is that IPE further takes the interactions between atomic clusters into account. More details on the IPE method and its integration into the Transformer architecture will be provided in the subsequent sections.

**Theorem 1** *Given two cluster $\sigma_i$ and $\sigma_j$ and their basis functions, there exists a set of invariant basis functions for the merged cluster $\sigma_{ij}$ to describe integrated cluster potentials $\tilde{E}_{ij}$.*

***Proof.*** A merged cluster $\sigma_{ij}$ can be represented as translating two clusters $\sigma_i$ and $\sigma_j$ such that their centered-atoms, i.e., atoms $i$ and $j$, overlap as shown in Fig. 1(b). All their neighbors are merged into a new cluster $\sigma_{ij}$, and the newly formed atomic base can be expressed as:

$$
\begin{aligned}
\tilde{A}_{ij,v} &= (A_{i,v} \quad A_{j,v}) \otimes (A_{i,v} \quad A_{j,v})^\top \\
&= \begin{pmatrix} A_{i,v}A_{i,v} & A_{i,v}A_{j,v} \\ A_{j,v}A_{i,v} & A_{j,v}A_{j,v} \end{pmatrix} \\
&= \begin{pmatrix} \sum_{m_1 m_2} \phi_v(\hat{r}_{im_1})\phi_v(\hat{r}_{im_2}) & \sum_{mn} \phi_v(\hat{r}_{im})\phi_v(\hat{r}_{jn}) \\ \sum_{mn} \phi_v(\hat{r}_{jn})\phi_v(\hat{r}_{im}) & \sum_{n_1 n_2} \phi_v(\hat{r}_{jn_1})\phi_v(\hat{r}_{jn_2}) \end{pmatrix}
\end{aligned}
\tag{6}
$$

with the product explicitly writing out. $m, n$ are the neighbor atom symbols of atoms $i, j$, respectively. $\otimes$ is the tensor product. We write the atomic base $A_{i,v}A_{j,v}$ as $A_{ij,v}$ (due to the permutational invariance, we have $A_{ij,v} = A_{ji,v}$) which still follows the *density trick*. Therefore, we could construct a new $A$-basis for merged cluster $\sigma_{ij}$ when taking product of $\tilde{A}_{ij,v}$:

$$
\begin{aligned}
\tilde{A}_{ij,\mathbf{v}} &= \tilde{A}_{ij,v_1} \odot \tilde{A}_{ij,v_2} \odot \cdots \odot \tilde{A}_{ij,v_\eta} \\
&= (A_{i,\mathbf{v}} \quad A_{j,\mathbf{v}}) \otimes (A_{i,\mathbf{v}} \quad A_{j,\mathbf{v}})^\top \\
&= \begin{pmatrix} \prod_{t=1}^{2\eta} A_{i,v_t} & \prod_{t=1}^{\eta} A_{ij,v_t} \\ \prod_{t=1}^{\eta} A_{ji,v_t} & \prod_{t=1}^{2\eta} A_{j,v_t} \end{pmatrix} \\
&= \begin{pmatrix} A_{i(i),\mathbf{v}} & A_{ij,\mathbf{v}} \\ A_{ji,\mathbf{v}} & A_{j(j),\mathbf{v}} \end{pmatrix}
\end{aligned}
\tag{7}
$$

where $2\eta = \epsilon$ and $\eta = 1, 2, \ldots$. $\tilde{A}_{ij,\mathbf{v}}$ could describe $(\epsilon + 1)$-body and $(\epsilon + 2)$-body expansion simultaneously in $\mathcal{O}(N)$. To be concrete, $A_{i(i),\mathbf{v}}$ contributes the $(\epsilon + 1)$-body expansion, while $A_{ij,\mathbf{v}}$ contributes the $(\epsilon + 2)$-body expansion due to cluster merging. For instance, when taking $\eta = 1$, $A_{i,\mathbf{v}}$ and $A_{i(i),\mathbf{v}}$ denote 2-body $(im)$ and 3-body expansion $(im_1 m_2)$ in original cluster $\sigma_i$, and $A_{ij,\mathbf{v}}$ denotes the 4-body expansion $(mijn)$ in merged cluster $\sigma_{ij}$, respectively. Further explanation can be found in the following Remark. $\tilde{A}_{ij,\mathbf{v}}$ exists when $\epsilon \geq 2$, which implies considering at least 4-body expansion within molecules. Then we could construct the corresponding matrix of $B$-basis $\tilde{B}_{ij,\mathbf{v}}$ following Equation 4 and represent the potential of cluster $\sigma_{ij}$ as:

$$
\tilde{E}_{ij} = \sum_\eta c_{ij,v} \tilde{B}_{ij,\mathbf{v}}
\tag{8}
$$

***Remark.*** A straightforward illustration can be drawn by setting $v = 1$ (Cartesian space) and $\eta = 1$. In this case, basis $B_{i(i)}$ could be interpreted as the sum of cosine value of the surrounding *angles* within cluster $\sigma_i$ [39, 43], i.e., $\sum_{m_1 m_2} \cos\theta_{im_1 m_2}$, which represents the 3-body contributions in $\mathcal{O}(N)$. Similarly, basis $B_{ij}$ could be treated as the sum of *proper dihedral angles* between two clusters $\sigma_i$ and $\sigma_j$ [49], i.e., $\sum_{mn} \cos\varphi_{mijn}$, which represents 4-body contributions in $\mathcal{O}(N)$. The basis $B_{ij}$ indeed serves as the contribution for torsion potential between two clusters in the original ACE, which primarily considers the contributions within a single cluster. The detailed proof could be found in Appendix B. As a result, incorporating $A_{ij}$ effectively enhances the representation of interatomic relations by capturing the interactions between clusters, thereby offering a possible way for designing a geometric Transformer architecture.

**Theorem 2 (Interactomic Positional Encoding (IPE))** *Given one molecule with $N$ atoms, there exists a positional encoding matrix $\mathbf{C}_\eta \in \mathbb{R}^{N \times N}$, which naturally describes the interatomic potentials. $\eta$ denotes the orders of body expansion in Equation 7. In particular, $\mathbf{C}_\eta$ is directly multiplied with Query and Key before scaling. e.g., softmax, serving as the positional encoding in Transformer:*

$$
\boldsymbol{\alpha} = (XW_Q)(XW_K)^\top \odot \mathbf{C}_\eta
\tag{9}
$$

*where $X \in \mathbb{R}^{N \times F}$ denotes the atomic features and $W \in \mathbb{R}^{F \times F}$ denotes the learnable matrix. $F$ is the hidden dimension. $Q, K$ represent Query and Key, respectively. $\odot$ is the Hadamard product.*

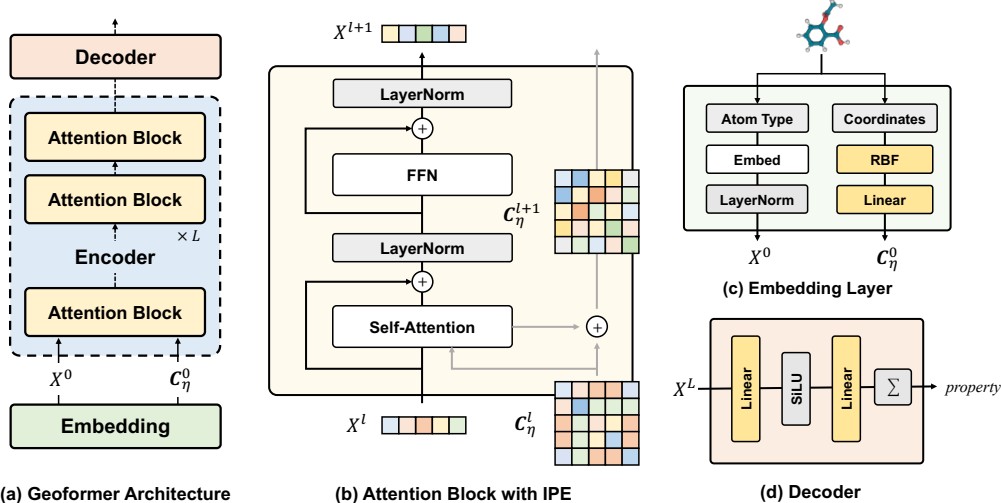

(a) Geoformer Architecture  (b) Attention Block with IPE  (c) Embedding Layer  (d) Decoder

Figure 2: **Geoformer Architecture.** Geoformer consists of (**c**) an Embedding layer; (**b**) an elaborate Encoder incorporating $L$ attention blocks with IPE $C_\eta$ to extract geometric features and capture complex interatomic relationships within the molecular structure. The extended Self-Attention module with $C_\eta$ is depicted in Fig. 1; (**d**) a lightweight Decoder for predicting molecular properties of interest, such as energy and HOMO-LUMO gap.

***Proof.*** As mentioned in Section 2 and Theorem 1, the basis functions $A_{i,\mathbf{v}}$ in Equation 3 could represent the local chemical environment of cluster $\sigma_i$ with *density trick*. Inspired by this, we first construct the $\boldsymbol{A}_{\mathbf{v}_\tau} = [\boldsymbol{A}_{\mathbf{1},\mathbf{v}_\tau}, \boldsymbol{A}_{\mathbf{2},\mathbf{v}_\tau}, \dots, \boldsymbol{A}_{\mathbf{N},\mathbf{v}_\tau}]^\top$ with integer $\tau = [1, \eta]$ and $\boldsymbol{A}_{i,v_\tau} = \prod_{t=1}^\tau A_{i,v_t}$. We then treat $\boldsymbol{A}_{\mathbf{v}_\tau}$ as the absolute positional encoding attached to the Query and Key, and attention matrix $\boldsymbol{\alpha}$ before scaling, e.g., softmax, could be modified as follows:

$$
\begin{aligned}
\boldsymbol{\alpha}_{\boldsymbol{\tau}} &= (XW_Q \circ \boldsymbol{A}_{\mathbf{v}_\tau})(XW_K \circ \boldsymbol{A}_{\mathbf{v}_\tau})^\top \\
&= (XW_Q)(XW_K)^\top \odot (\boldsymbol{A}_{\mathbf{v}_\tau} \boldsymbol{A}_{\mathbf{v}_\tau}^\top) \\
&= (XW_Q)(XW_K)^\top \odot \sum_{\mathbf{v}_\tau} \tilde{\boldsymbol{A}}_{\mathbf{v}_\tau}
\end{aligned}
\tag{10}
$$

where $\circ$ is the entry-wise Kronecker Product. The detailed proof could be found in Appendix B. Instructed by [10, 21], we add Clebsch-Gordan coefficients $C_{\mathbf{v}_\tau}$ to ensure the rotationally invariance, and therefore obtain $\tilde{\boldsymbol{B}}_{\mathbf{v}_\tau} = \sum_{\mathbf{v}'_\tau} C_{\mathbf{v}\mathbf{v}'_\tau} \tilde{\boldsymbol{A}}_{\mathbf{v}'_\tau}$. Then we can write the linear expansion:

$$
\boldsymbol{\alpha} = (XW_Q)(XW_K)^\top \odot \left( \sum_{\tau=1}^\eta W_{\tilde{B}} \tilde{\boldsymbol{B}}_{\mathbf{v}_\tau} \right)
\tag{11}
$$

where $W_{\tilde{B}}$ is a learnable weight matrix denoting $c$-coefficients, which is similar to the message construction in MACE [2]. Such operation could be done by a tensor broadcast. Eventually, our interatomic positional encoding $C_\eta$ could be represented as:

$$
\boldsymbol{C}_\eta = \sum_{\tau=1}^\eta W_{\tilde{B}} \tilde{\boldsymbol{B}}_{\mathbf{v}_\tau}
\tag{12}
$$

We can further modify the Equation 11 as Equation 9 and complete the proof.

### 3.2 Geoformer: Geometric Transformer for Molecules

**Overall Design.** The comprehensive structure of Geoformer is depicted in Figure 2. This design comprises an Embedding layer, an elaborate Encoder for extracting geometric features, and a lightweight Decoder for predicting molecular properties. Within the Geoformer's Encoder, $L$ attention blocks are integrated, each having hidden dimension $F$ and employing the proposed interatomic positional encoding $C_\eta$. Notably, $C_\eta$ is updated by the atomic features $X \in \mathbb{R}^{N \times F}$ within each block. Geoformer takes the atom type $Z \in \mathbb{R}^N$ and atomic coordinates $R \in \mathbb{R}^{N \times 3}$ from one molecule with $N$ atoms as inputs, and produces the corresponding molecular properties as outputs.

**Embedding Layer.** The Embedding layer maps the atom type $Z$ to $X^0 \in \mathbb{R}^{N \times F}$:

$$X^0 = \text{LayerNorm}\left(\text{embed}(Z)\right) \tag{13}$$

and initialize the IPE $\boldsymbol{C}_\eta^0 \in \mathbb{R}^{N \times N \times F}$ with 2-body expansion, i.e., radial basis functions (RBF) [45]:

$$g_k(\hat{R}) = \phi(\|\hat{R}\|) \cdot \exp\left(-\beta_k \left(\exp\left(-\|\hat{R}\|\right) - \mu_k\right)^2\right) \tag{14}$$

$$\boldsymbol{C}_\eta^0 = \mathbf{g}(\hat{R})W_{\text{RBF}} \tag{15}$$

where $\hat{R} \in \mathbb{R}^{N \times N \times 3}$ denotes the relative position between two atoms, $\|\cdot\|$ is the vector norm, $\beta_k$, $\mu_k$ are optional learnable parameters that specify center and width of $g_k(\hat{R})$, and $\phi(\cdot)$ is a smooth cosine cutoff function. $\mathbf{g}(\hat{R}) = [g_1(\hat{R}), \ldots, g_K(\hat{R})]^\top \in \mathbb{R}^{N \times N \times K}$ is composed of the values of $K$ radial basis functions. $W_{\text{RBF}} \in \mathbb{R}^{K \times F}$ is a learnable matrix mapping basis functions to the hidden size.

**Attention block with IPE.** The extended attention block is illustrated in Fig. 1(a). In contrast to the traditional Transformer, a learnable IPE matrix $\boldsymbol{C}_\eta$ is introduced to each attention block. Following TorchMD-NET [43], the softmax function is substituted with the SiLU activation to enhance accuracy, and the attention weight is scaled by a smooth cutoff:

$$A(X^l) = \text{SiLU}\left(\sum_F \left(\left((X^l W_Q^l) * (X^l W_K^l)^\top\right) \odot \boldsymbol{C}_\eta^l\right)\right) \cdot \phi(\|\hat{R}\|) \tag{16}$$

where $*$ denotes the batched tensor product, i.e., $(X^l W_Q^l) * (X^l W_K^l)^\top \in \mathbb{R}^{N \times N \times F}$. $l$ indicates $l$-th attention block. In Fig. 1, the mask operation corresponds to the implementation of an alternative attention mask. It effectively filters out atoms with excessive distance, concentrates attention and improves the performance of the attention mechanism [5]. Then we produce the weighted values per atoms after self-attention to update $\boldsymbol{C}_\eta^l$:

$$\text{Attn}_V(X^l) = A(X^l) \odot X^l W_V^l \tag{17}$$

Then under the instruction of Equation 7, we construct the $A$-basis for all merged cluster:

$$\boldsymbol{A}_{\mathbf{v}_\tau}^l = \prod_{t=1}^\tau \sum_j \text{Attn}_V(X^l) W_{Attn}^l Y_{l^*,m}(\hat{R}/\|\hat{R}\|) \tag{18}$$

$$\tilde{\boldsymbol{A}}_{\mathbf{v}_\tau}^l = (\boldsymbol{A}_{\mathbf{v}_\tau}^l W_{\tilde{A}_{\mathbf{v}_t}^{(1)}}^l) \otimes (\boldsymbol{A}_{\mathbf{v}_\tau}^l W_{\tilde{A}_{\mathbf{v}_t}^{(2)}}^l)^\top \tag{19}$$

where $W_{Attn}^l$ is a learnable matrix, $Y_{l^*,m^*}(\hat{R}/\|\hat{R}\|)$ denotes the spherical harmonics with order $l^*$ and degree $m^*$. $W_{\tilde{A}_{\mathbf{v}_t}^{(1)}}^l$ and $W_{\tilde{A}_{\mathbf{v}_t}^{(2)}}^l$ are two learnable matrix *without bias* to ensure equivariance [39]. We could further construct a new form of residual IPE within each block following Equation 8 and 11:

$$\delta \boldsymbol{C}_\eta^l = \sum_{\tau=1}^\eta W_{\tilde{B}}^l \sum_{\mathbf{v}_\tau'} C_{\mathbf{v}\mathbf{v}_\tau'} \tilde{\boldsymbol{A}}_{\mathbf{v}_\tau'} \tag{20}$$

where $W_{\tilde{B}}^l$ is a learnable matrix. Finally we apply the residual connection to compute IPE for the next block:

$$\boldsymbol{C}_\eta^{l+1} = \text{SiLU}\left(\boldsymbol{C}_\eta^l W_{\boldsymbol{C}}^l\right) \odot \delta \boldsymbol{C}_\eta^l + \boldsymbol{C}_\eta^l \tag{21}$$

where $W_{\boldsymbol{C}}^l$ is a learnable matrix and $\text{SiLU}(\boldsymbol{C}_\eta^l W_{\boldsymbol{C}}^l)$ plays a role of gated filter. The update of atomic features still follows the traditional procedures:

$$\text{Attn}(X^l) = \sum_j \text{Attn}_V(X^l) + X^l \tag{22}$$

$$\text{FFN}(X^l) = \text{SiLU}(X^l W_1^l)W_2^l + X^l \tag{23}$$

where $\sum_j$ denotes the sum of atomic values weighted by the attention score. $W_1^l$ and $W_2^l$ are two learnable matrix in feed-forward layer. The output of each step is fed into a Layer Normalization (LayerNorm). It is important to emphasize that while the theoretical derivations of the Geoformer architecture above may appear complex, in practice, the model utilizes simplified settings to reduce computational complexity. Specifically, we employ $\eta = 1$ for body expansion and $l^* = 1$ for spherical harmonics. It streamlines the complex tensor contraction and Clebsch-Gordan product calculations, making the model more efficient and easier to implement. Despite this, Geoformer achieves or

surpasses state-of-the-art prediction results on several benchmark datasets, as demonstrated in the subsequent section.

**Decoder.** The Geoformer utilizes a lightweight Decoder, as depicted in Fig. 2(d). It consists of a two linear layers with SiLU activation and an aggregation module $\sum$ to predict the specific molecular property. The lightweight Decoder ensures that the Geoformer remains computationally efficient while maintaining its ability to accurately predict molecular properties based on the geometric features captured by the Encoder. More details on Decoder for specific properties are shown in Appendix G.

## 4 Experiments

### 4.1 Experimental Setup

Geoformer is evaluated on both QM9 dataset [35] that consists of 12 molecular properties and a large-scale Molecule3D dataset [51] derived from PubChemQC [32] with ground-state structures and the corresponding properties calculated at DFT level. All results are measured by mean absolute error (MAE) on test sets and baseline results are directly taken from the corresponding papers. All models are trained using the AdamW optimizer, and we use the learning rate decay if the validation loss stops decreasing. We also adopt the early stopping strategy to prevent over-fitting. The optimal hyperparameters such as learning rate and batch size are selected on validation sets. More detailed hyperparameters setting for Geoformer are provided in Appendix Table 4.

### 4.2 QM9

Table 1: Mean absolute errors (MAE) of 12 kinds of molecular properties on QM9 compared with state-of-the-art algorithms. The best one in each category is highlighted in **bold**.

| Target
Unit | $\mu$
$mD$ | $\alpha$
$ma_0^3$ | $\epsilon_{HOMO}$
$meV$ | $\epsilon_{LUMO}$
$meV$ | $\Delta\epsilon$
$meV$ | $\langle R^2 \rangle$
$ma_0^2$ | $ZPVE$
$meV$ | $U_0$
$meV$ | $U$
$meV$ | $H$
$meV$ | $G$
$meV$ | $C_v$
$\frac{mcal}{mol\ K}$ |
|---|---|---|---|---|---|---|---|---|---|---|---|---|
| NMP [14] | 30 | 92 | 43 | 38 | 69 | 180 | 1.50 | 20 | 20 | 17 | 19 | 40 |
| SchNet [38] | 33 | 235 | 41 | 34 | 63 | 73 | 1.70 | 14 | 19 | 14 | 14 | 33 |
| Cormorant [1] | 38 | 85 | 34 | 38 | 61 | 961 | 2.03 | 22 | 21 | 21 | 20 | 26 |
| LieConv [11] | 32 | 84 | 30 | 25 | 49 | 800 | 2.28 | 19 | 19 | 24 | 22 | 38 |
| DimeNet++ [13] | 30 | 44 | 25 | 20 | 33 | 331 | 1.21 | 6.32 | 6.28 | 6.53 | 7.56 | 23 |
| EGNN [37] | 29 | 71 | 29 | 25 | 48 | 106 | 1.55 | 11 | 12 | 12 | 12 | 31 |
| PaiNN [39] | 12 | 45 | 28 | 20 | 46 | 66 | 1.28 | 5.85 | 5.83 | 5.98 | 7.35 | 24 |
| TorchMD-NET [42] | 11 | 59 | 20 | 18 | 36 | 33 | 1.84 | 6.15 | 6.38 | 6.16 | 7.62 | 26 |
| GNS + NoisyNode [15] | 25 | 52 | 20 | 19 | 29 | 700 | 1.16 | 7.30 | 7.57 | 7.43 | 8.30 | 25 |
| SphereNet [25] | 25 | 45 | 23 | 19 | 31 | 268 | **1.12** | 6.26 | 6.36 | 6.33 | 7.78 | **22** |
| SEGNN [4] | 23 | 60 | 24 | 21 | 42 | 660 | 1.62 | 15 | 13 | 16 | 15 | 31 |
| EQGAT [23] | 11 | 53 | 20 | 16 | 32 | 382 | 2.00 | 25 | 25 | 24 | 23 | 24 |
| PaxNet [54] | 11 | 45 | 23 | 19 | 31 | 249 | 1.17 | 5.90 | 5.92 | 6.04 | 7.14 | 23 |
| ComENet [47] | 25 | 45 | 23 | 20 | 32 | 259 | 1.20 | 6.59 | 6.82 | 6.86 | 7.98 | 24 |
| Equiformer [24] | 11 | 46 | **15** | **14** | **30** | 251 | 1.26 | 6.59 | 6.74 | 6.63 | 7.63 | 23 |
| AMP [44] | 12 | 67 | 26 | 23 | 45 | 93 | 4.10 | 11.3 | 11.4 | 11.3 | 12.4 | 32 |
| Molformer [50] | 28 | 41 | 25 | 26 | 39 | 350 | 2.05 | 7.52 | 7.46 | 7.38 | 8.11 | 25 |
| GeoT [22] | 29.7 | 52.7 | 25.0 | 20.2 | 43.9 | 300.8 | 1.73 | 11.1 | 11.7 | 11.3 | 11.7 | 27.6 |
| Geometric Transformer [8] | 26.4 | 51 | 27.5 | 20.4 | 36.1 | 157 | 1.24 | 7.35 | 7.55 | 7.73 | 8.21 | 28.0 |
| Transformer-M [28] | 37 | 41 | 17.5 | 16.2 | 27.4 | 75 | 1.18 | 9.37 | 9.41 | 9.39 | 9.63 | **22** |
| Geoformer | **10** | **40** | 18.4 | 15.4 | 33.8 | **27.5** | 1.28 | **4.43** | **4.41** | **4.39** | **6.13** | **22** |

QM9 dataset consists of 130,831 small organic molecules with up to 9 heavy atoms. Each molecule is associated with 12 targets covering its energetic, electronic, and thermodynamic properties. We randomly split them in to 110,000 samples for training, 10,000 samples for validation and the remains for testing following the prior work [43]. The evaluation results on QM9 are shown in Table 1 with the upper section displaying EGNNs and the lower showcasing Transformer-based methods. When compared with other SoTA methods, Geoformer achieves the state-of-the-art results on 8 kinds of properties and shows comparable results on the remaining properties, which underscores that our IPE can help Transformers learn useful positional information to better model molecular structures. Specifically, we conduct a comparison between Geoformer and the previously best-performing Transformer-based model, Transformer-M, which incorporates pairwise distances as PE. As Transformer-M has been pretrained on the large-scale PCQM4Mv2 dataset [17], targeting the prediction of HOMO-LUMO gaps, it exhibits comparable performance in terms of related properties. Nevertheless, a noticeable performance disparity persists when evaluating other properties

in comparison to the EGNNs. By integrating more directional information beyond distances into Geoformer, it significantly surpasses Transformer-M in 9 kinds of properties without pretraining.

## 4.3 Molecule3D

The Molecule3D dataset consists of 3,899,647 molecules, each with the corresponding ground-state structures and quantum properties calculated by Density Functional Theory (DFT). The dataset is split into train, validation, and test set with the ratio of 6:2:2. The official GitHub repository of Molecule3D provides both random and scaffold splits, which are both employed in our experiments. The random split ensures that the training, validation, and test sets are sampled from the same distribution, while the scaffold split introduces a distribution shift among different subsets. We focus our analysis on the prediction of the HOMO-LUMO gap, in comparison with ComENet [47]. Table 2 displays the results of our experiments on Molecule3D,

Table 2: Mean absolute errors (MAE) of HOMO-LUMO gap (eV) on Molecule3D test set for both random and scaffold splits compared with state-of-the-art algorithms.

| Model | Random | Scaffold |
|---|---|---|
| GIN-Virtual [18] | 0.1036 | 0.2371 |
| SchNet [38] | 0.0428 | 0.1511 |
| DimeNet++ [13] | 0.0306 | 0.1214 |
| SphereNet [25] | 0.0301 | 0.1182 |
| ComENet [47] | 0.0326 | 0.1273 |
| Geoformer | **0.0202** | **0.1135** |

indicating that Geoformer achieved a reduction of 32.56% and 3.98% in test MAE on the random and scaffold splits, respectively. These results highlight the superiority of our approach compared to invariant GNNs on a large-scale dataset.

## 4.4 Analysis on Interatomic Positional Encoding

The exceptional performance demonstrated by our Geoformer serves as an evidence that Interatomic Positional Encoding (IPE) can effectively guide Transformers in modeling the geometry of molecular systems. In order to analyze the distinctions between our learned IPE and other PEs that solely utilize pairwise distances, we visualize the positional encoding for different molecules by IPE and other PEs, respectively. As shown in Fig. 3, IPE exhibits different positional encoding information compared with PEs that solely utilize pairwise distances. Specifically, for some positions that exhibit strong signals shown in the PEs only with distances, IPE further enhances such signals and shows significant distinction from background signals.

## 4.5 Ablation Study

To verify the effectiveness of our IPE, we conduct comprehensive ablation studies with model variants on four properties $U_0$, $U$, $H$ and $G$ in QM9. First, we remove the residual connection for IPE, which leads to a Transformer that exclusively encodes pairs of distances using the initial non-updated IPE (**Non-updated**

Table 3: Ablation study on four properties $U_0$, $U$, $H$ and $G$ in QM9 test set for model variants. The best one in each property is highlighted in bold.

| Property | Non-updated IPE | Addition | Geoformer |
|---|---|---|---|
| $U_0$ | 5.63 | 7.62 | **4.43** |
| $U$ | 5.87 | 7.66 | **4.41** |
| $H$ | 6.21 | 7.93 | **4.39** |
| $G$ | 7.24 | 9.01 | **6.13** |

**IPE**). More specifically, the RBF feature is constructed as Equation 14, where $\mathbf{g}(\hat{R}) = [g_1(\hat{R}), \ldots, g_K(\hat{R})]^\top \in \mathbb{R}^{N \times N \times K}$ is composed of the values of $K$ radial basis functions and the PE undergoes a transformation in the $l$-th layer through a distinct linear layer that is not shared across layers with the form:

$$\boldsymbol{C}_\eta^l = \mathbf{g}(\hat{R}) W_{\text{RBF}}^l \tag{24}$$

The remaining operations are consistent with those found in the original Geoformer implementation. This should be highly similar to the previous Transformer, albeit with a slightly different pairwise distance encoding approach.

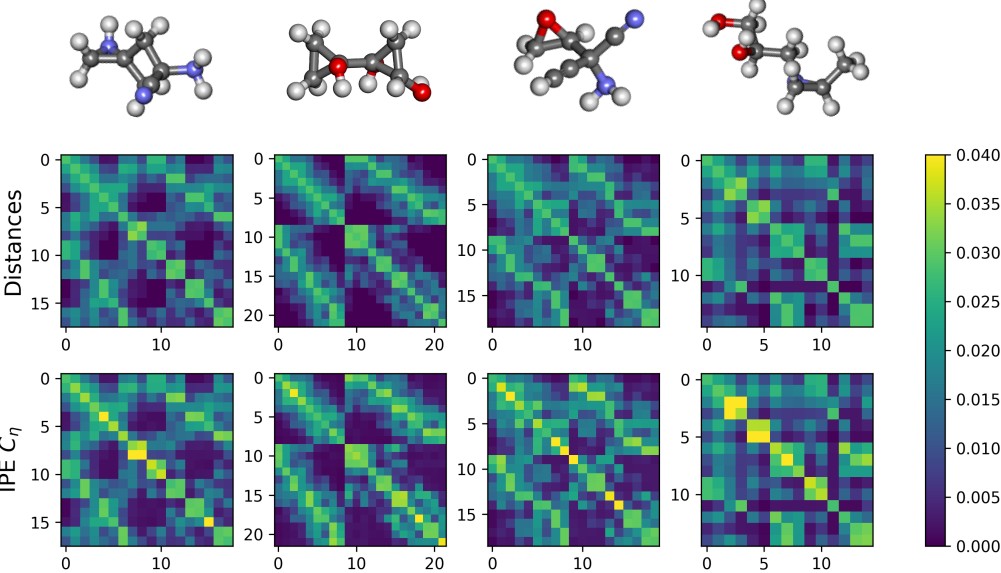

Figure 3: Visualization of IPE $C_\eta$ on molecules GDB65488, GDB101712, GDB87153 and GDB56373 in QM9 test set. Other PEs (first row) only encode pairwise distances by RBF, and the learned IPE $C_\eta$ (second row) encodes different positional information beyond pairwise distance. In some atomic positions, IPE effectively emphasizes the atomic relationships, providing a more comprehensive representation of the underlying geometry within the molecular structure. This enhanced encoding of geometric information enables the Transformer-based model to capture the intricate interactions between atoms and better predict molecular properties.

Second, we discover that the way of incorporating IPE into the attention weights is crucial. In our derivation (Eq. 10), multiplication emerges as a natural approach, whereas the addition of PEs as attention bias prevails in the previous Transformer. To verify the effectiveness of our design, we also introduce our IPEs into the Transformer in an additive manner (**Addition**), which we replace multiplication to addition as follows:

$$A(X^l) = \text{SiLU}\left(\sum_F \left(\left((X^l W_Q^l) * (X^l W_K^l)^\top\right) + C_\eta^l\right)\right) \cdot \phi(\|\hat{R}\|) \tag{25}$$

where $C_\eta^l$ updated in the same way as the original Geoformer, only $\odot$ has been replaced with $+$. As show in Table 3, the performance of Geoformer without the updated IPE is worse than the original version, which shows that the additional directional information is important.

The performance also drops when using the addition of PEs as the attention bias, which demonstrates the multiplication is a proper way to incorporate IPE into Transformers. Furthermore, the results for both variants outperform Transformer-M, which uses pairwise distance information as the attention bias. This observation further highlights the significance of these components to the overall performance gains.

## 5 Related Work

### 5.1 Positional Encoding in Transformers

The Transformer architecture contains a series of Transformer blocks as well as a positional encoding (PE) to model the sequence data. Since the self-attention modules in Transformer blocks are invariant to the sequence orders, the positional encoding plays an essential role in injecting positional information from sequences into the Transformer. Recent PEs can be categorized into absolute PE and relative PE. The original Transformer model incorporates absolute PE by directly adding the positional encoding to the tokens [46]. Although absolute PE has been demonstrated to approximate any continuous sequence-to-sequence functions [53], it tends to exhibit inferior generalization capabilities

in comparison to relative PE, particularly when dealing with longer sequences[33]. The relative PE further considers the pairwise relationship between two tokens. Shaw [41], T5 [34], DeBERTa [16], and Transformer-XL [9] have developed various relative PE approaches for parameterizing relative positions. The success of relative PE in natural language processing has inspired its applications in other domains. For instance, the Swin Transformer [27] employs relative PE to model relationships between image patches, while Graphormer[52] utilizes both absolute PE (centrality encoding) and relative PE (spatial encoding and edge encoding) to model the graph topology. Recently, Transformer-M [28] and Uni-Mol [55] have incorporated pairwise distances as relative positional encoding to capture positional information within 3D space. TorchMD-Net [43] included the radial basis functions (RBF) as distance filter to the attention matrix. MUformer [19] further extended the distance filter by incorporating additional 2D structural information, resulting in improved performance and applicability to molecule 2D-3D co-generation. Several works have further explored the integration of different types of positional encoding in Transformers. GPS [36] offers an comprehensive overview of the available PE methods employed in graph Transformers. The MAT [29] and R-MAT [30] approaches methodologically introduce inter-atomic distances and chemical bond information as domain-specific inductive biases in Transformer architecture. Molformer [50] employs Adaptive PE to model molecules of varying sizes; GeoT [22] forgoes softmax in favor of distance matrices as a scaling factor, and the application of multiplication for PE has been previously tried in the Geometric Transformer [8]. In this study, our objective is to develop a relative PE for modeling molecular geometry. Drawing upon the atomic cluster expansion theory, which will be discussed in Section 2, we derive a rotation-invariant relative PE that incorporates additional positional information beyond pairwise distances.

## 5.2 Geometric Deep Learning for Molecules

Geometric deep learning (GDL) has emerged as a promising approach to modeling molecular geometry and predicting the properties of molecules, playing an important role in fields such as drug discovery, materials science, and computational chemistry. By leveraging the inherent geometric structures and incorporating symmetry in architecture design, GDL approaches offer an effective and efficient representation for molecules. Invariant and equivariant graph neural networks (EGNNs) are representative methods in GDL. SchNet [38], DimeNet++ [13], GemNet [12], SphereNet [25], ComENet [47] gradually explicitly incorporate more geometric information including distances, angles and dihedrals. Some works like PaiNN [39], TorchMD-Net [43], ViSNet [49] adopt vector embedding and implicitly extract the above geometric information with lower consumption. Another mainstream approach such as NequIP [3], MACE [2], Allegro [31] and Equiformer [24] guarantee equivariance through group representation theory, which can achieve higher accuracy leveraging high-order geometric tensors.

## 6 Discussion

In this paper, we propose a novel Transformer architecture in the field of molecular modeling. This innovative approach, incorporating the novel Interatomic Positional Encoding (IPE), effectively captures complex geometric information and interatomic relations beyond pairwise distances embeded in the molecular structures. The extensive results on QM9 and Molecule3D dataset elucidate the capability of Geoformer compared with Transformers and EGNNs. Further research can explore its applicability to a broader range of systems such as materials and polymers. Moreover, the concept Interatomic Positional Encoding may inspire the development of more advanced encoding schemes in Transformers.

**Limitation and Societal Impacts:** Like other Transformer-based architectures, Geoformer suffers from common training instabilities, necessitating the use of a relatively small learning rate during the training process. Effective molecular geometry modeling, as provided by the Geoformer, significantly benefits the materials science and pharmaceutical industries. However, it is essential to acknowledge that the same technology could be misused for illicit activities, such as the manufacturing of illegal drugs or the development of biochemical weapons.

## Acknowledgments and Disclosure of Funding

We thank the reviewers for their valuable comments. Yusong Wang and Nanning Zheng were supported in part by NSFC under grant No. 62088102.

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
