# A  Annotations

Table 1: Glossary of notations

| Notation | Description |
|---|---|
| $i; j; k; m; n$ | Symbol of atoms |
| $N$ | Number of neighbors |
| $N(i)$ | Neighborhood of node $i$ |
| $\sigma_i, \sigma_j, \sigma_{ij}$ | Cluster $i, j$ and their merged cluster |
| $\upsilon$ | Polynomial degree |
| $\phi_\upsilon(\cdot)$ | Basis function |
| $c_\upsilon$ | Expansion coefficients |
| $E_i$ | Potential of cluster $\sigma_i$ |
| $A_\upsilon$ | Atomic base for single cluster |
| $\epsilon$ | Order of expansion for single cluster |
| $A_\mathbf{v}$ | $A$-basis for single cluster |
| $B_\mathbf{v}$ | $B$-basis for single cluster |
| $\tilde{A}_\upsilon$ | Atomic base for merged cluster |
| $\eta$ | Order of expansion for merged cluster |
| $\tilde{A}_\mathbf{v}$ | $A$-basis for merged cluster |
| $\tilde{B}_\mathbf{v}$ | $B$-basis for merged cluster |
| $\tilde{E}_{ij}$ | Potential of merged cluster $\sigma_{ij}$ |
| $\tau$ | Specific expansion order |
| $\boldsymbol{A_{\mathbf{v}_\tau}}$ | Matrix of $A$-basis for single cluster on order $\tau$ |
| $\boldsymbol{\tilde{A}_{\mathbf{v}_\tau}}$ | Matrix of $A$-basis for merged cluster on order $\tau$ |
| $\boldsymbol{\tilde{B}_{\mathbf{v}_\tau}}$ | Matrix of $B$-basis for merged cluster on order $\tau$ |
| $\boldsymbol{\alpha}$ | Matrix of attention score |
| $\boldsymbol{C}_\eta$ | Interatomic Positional Encoding |
| $l; L$ | Index of blocks; Number of blocks |
| $F$ | Hidden dimension |
| $X$ | Atomic features |
| $\sum \cdot$ | Reduce function |
| $\|\cdot\|$ | Vector Norm |
| $Z$ | Atom type |
| $\hat{R}$ | Matrix of relative positions |
| $g(\cdot), \mathrm{g}(\cdot)$ | Radial basis functions |
| $\mathrm{embed}(\cdot)$ | Embedding operation |
| $\mathrm{LayerNorm}$ | Layer normalization operation |
| $\langle \cdot, \cdot \rangle$ | Dot (Inner) product |
| $\mathrm{SiLU}(\cdot)$ | SiLU activation function |
| $W$ | Learnable weight matrix |
| $Y_{l^*,m}(\cdot)$ | Spherical harmonics with order $l^*$ and degree $m$ |
| $C_{\mathbf{v}\mathbf{v}_\tau}$ | Clebsch-Gordan coefficients |
| $\mathrm{Attn}_V(\cdot)$ | Attention values weighed by attention scores before sum |
| $\mathrm{Attn}(\cdot)$ | Attention values weighed by attention scores after sum |
| $\mathrm{FFN}(\cdot)$ | Feed-forward layer |
| $\odot$ | Hadamard product |
| $\otimes$ | Kronecker product or Tensor product |
| $\circ$ | Entry-wise Kronecker product |
| $*$ | Batched tensor product |
| $\top$ | Matrix transpose |
| $\delta$ | Residual |

# B Detailed Proof of Interatomic Positional Encoding

By setting $\upsilon = 1$ and $\eta = 1$, we have the basis function $\phi_\upsilon(\cdot)$ to be its unit vector in Cartesian space (setting order $l^* = 1$ and degree $m = 3$ in spherical harmonics). Consequently, the atom base could be represented as:

$$A_{i,\upsilon=1} = \sum_{m \in N(i)} \left( \frac{\hat{r}_{im}}{\|\hat{r}_{im}\|} \right) \tag{1}$$

The components of $B$-basis for the merged cluster $\sigma_{ij}$ can be written as:

$$
\begin{aligned}
B_{i(i),\mathbf{v}} &= \sum_{\mathbf{v}'} C_{\mathbf{v}\mathbf{v}'} A_{i(i),\mathbf{v}'} \\
&= \sum_{m^*=-l^*}^{l^*} (-1)^{m^*} A_{i,l^*m^*} A_{i,l^*-m^*} \\
&= \sum_{m_1} \sum_{m_2} \left\langle \frac{\hat{r}_{im_1}}{\|\hat{r}_{im_1}\|}, \frac{\hat{r}_{im_2}}{\|\hat{r}_{im_2}\|} \right\rangle \\
&= \sum_{m_1} \sum_{m_2} \langle \hat{u}_{im_1}, \hat{u}_{im_2} \rangle \\
&= \sum_{m_1 m_2} \cos \theta_{im_1 m_2}
\end{aligned}
\tag{2}
$$

where $\langle \cdot, \cdot \rangle$ denotes the inner product. We treat $A_i$ as complex spherical harmonics [4]. $\cos \theta_{im_1 m_2}$ represents the cosine value of *angles* formed by atom $i, m_1, m_2$. We simplify the $\hat{r}_{im}/\|\hat{r}_{im}\|$ as $\hat{u}_{im}$.

$$
\begin{aligned}
B_{ij,\mathbf{v}} &= \sum_{\mathbf{v}'} C_{\mathbf{v}\mathbf{v}'} A_{ij,\mathbf{v}'} \\
&= \sum_{m^*=-l^*}^{l^*} (-1)^{m^*} A_{i,l^*m^*} A_{j,l^*-m^*} \\
&= \sum_m \sum_n \left( \hat{w}_{ij} + \langle A_{i,\upsilon=1}, \hat{u}_{ij} \rangle \cdot \vec{u}_{ij} \right) \left( \hat{w}_{ji} + \langle A_{j,\upsilon=1}, \hat{u}_{ji} \rangle \cdot \vec{u}_{ji} \right) \\
&\propto \sum_m \sum_n \langle \hat{w}_{im}, \hat{w}_{jn} \rangle + C \\
&= \sum_{mn} \cos \varphi_{mijn} + C
\end{aligned}
\tag{3}
$$

with $C$ denoting an vector offset and could be integrated into neural network. $\hat{w}_{im}$ denotes the vectors vertical to the intersection line between two planes for calculating the $\cos \varphi_{mijn}$ with $\langle \hat{w}_{ij}, \hat{u}_{ij} \rangle = 0$. The inner product of $\hat{w}_{im}$ and $\hat{w}_{jn}$ represents the cosine value of *proper dihedral angles* formed by plane $i, j, n$ and plane $i, j, m$.

In a nutshell, we make a straightforward illustration of the relation between our basis functions and geometric information within molecular structures.

**Theorem 3** *Given matrix $A_{m \times n}, B_{n \times k}, C_{l \times p}, D_{p \times q}$, we have*

$$A \otimes C \in \mathbb{R}^{ml \times np} \tag{4}$$

$$(A \otimes B)^\top = A^\top \otimes B^\top \tag{5}$$

$$(A \otimes C)(B \otimes D) = (AB) \otimes (CD) \tag{6}$$

*where $\otimes$ denotes the Kronecker Product.*

---

[4]https://en.wikipedia.org/wiki/Table_of_spherical_harmonics

Given Theorem 3, the proof of Interatomic Positional Encoding (IPE) could be written as at length:

$$
\begin{aligned}
\boldsymbol{\alpha_\tau} &= (XW_Q \circ \boldsymbol{A_{v_\tau}})(XW_K \circ \boldsymbol{A_{v_\tau}})^\top \\
&= \begin{pmatrix} x_1 W_Q \otimes \boldsymbol{A_{1,v_\tau}} \\ x_2 W_Q \otimes \boldsymbol{A_{2,v_\tau}} \\ \vdots \\ x_N W_Q \otimes \boldsymbol{A_{N,v_\tau}} \end{pmatrix} \cdot \begin{pmatrix} x_1 W_K \otimes \boldsymbol{A_{1,v_\tau}} \\ x_2 W_K \otimes \boldsymbol{A_{2,v_\tau}} \\ \vdots \\ x_N W_K \otimes \boldsymbol{A_{N,v_\tau}} \end{pmatrix}^\top \\
&= \begin{pmatrix} x_1 W_Q \otimes \boldsymbol{A_{1,v_\tau}} \\ x_2 W_Q \otimes \boldsymbol{A_{2,v_\tau}} \\ \vdots \\ x_N W_Q \otimes \boldsymbol{A_{N,v_\tau}} \end{pmatrix} \cdot \begin{pmatrix} (x_1 W_K)^\top \otimes (\boldsymbol{A_{1,v_\tau}})^\top \\ (x_2 W_K)^\top \otimes (\boldsymbol{A_{2,v_\tau}})^\top \\ \vdots \\ (x_N W_K)^\top \otimes (\boldsymbol{A_{N,v_\tau}})^\top \end{pmatrix} \\
&= \begin{pmatrix} (x_1 W_Q)(x_1 W_K)^\top \cdot (\boldsymbol{A_{1,v_\tau}} \boldsymbol{A_{1,v_\tau}^\top}) \\ (x_2 W_Q)(x_2 W_K)^\top \cdot (\boldsymbol{A_{2,v_\tau}} \boldsymbol{A_{2,v_\tau}^\top}) \\ \vdots \\ (x_N W_Q)(x_N W_K)^\top \cdot (\boldsymbol{A_{N,v_\tau}} \boldsymbol{A_{N,v_\tau}^\top}) \end{pmatrix} \\
&= (XW_Q)(XW_K)^\top \odot (\boldsymbol{A_{v_\tau}} \boldsymbol{A_{v_\tau}^\top}) \\
&= (XW_Q)(XW_K)^\top \odot \sum_{\mathbf{v_\tau}} \boldsymbol{\tilde{A}_{v_\tau}}
\end{aligned}
\tag{7}
$$

where $\circ$ is the entry-wise Kronecker Product and $\otimes$ is Kronecker Product.

# C Experiments on the MD17 Dataset

Geoformer is also evaluated on the MD17 dataset [6, 40, 7]. The MD17 dataset comprises MD trajectories of 7 small organic molecules, and the number of conformations for each molecule varies between 133,700 and 993,237. The objective is to predict potential energy and forces. Geoformer is trained in scenarios with limited data, using only 950 samples for training, 50 for validation, and the remaining for testing following the previous studies [39, 43]. As demonstrated in Appendix Table 2, it is noteworthy that Geoformer surpasses a series of EGNNs by achieving the lowest mean absolute errors (MAE) for predicted energy and forces on the majority of molecules.

Table 2: Mean absolute errors (MAE) of energy (kcal/mol) and force (kcal/mol/Å) for 7 small organic molecules on MD17 compared with state-of-the-art EGNNs. The best one in each category is highlighted in bold.

| Molecule | | PaiNN [39] | TorchMD-Net [43] | GemNet [12] | NequIP [3] | Geoformer |
|---|---|---|---|---|---|---|
| Aspirin | energy | 0.167 | 0.123 | - | 0.131 | **0.116** |
| | forces | 0.338 | 0.253 | 0.217 | 0.184 | **0.169** |
| Ethanol | energy | 0.064 | 0.052 | - | **0.051** | **0.051** |
| | forces | 0.224 | 0.109 | 0.085 | 0.071 | **0.063** |
| Malondialdehyde | energy | 0.091 | 0.077 | - | 0.076 | **0.074** |
| | forces | 0.319 | 0.169 | 0.155 | 0.129 | **0.115** |
| Naphthalene | energy | 0.116 | **0.085** | - | 0.113 | 0.087 |
| | forces | 0.077 | 0.061 | 0.051 | **0.039** | 0.043 |
| Salicylic Acid | energy | 0.116 | **0.093** | - | 0.106 | **0.093** |
| | forces | 0.195 | 0.129 | 0.125 | 0.090 | **0.088** |
| Toluene | energy | 0.095 | **0.074** | - | 0.092 | 0.078 |
| | forces | 0.094 | 0.067 | 0.060 | 0.046 | **0.044** |
| Uracil | energy | 0.106 | **0.095** | - | 0.104 | **0.095** |
| | forces | 0.139 | 0.095 | 0.097 | 0.076 | **0.066** |

# D Computational Efficiency

We conducted experiments about computational efficiency of Geoformer on a NVIDIA V100 GPU and compared the model sizes and training times reported in Equiformer [24] and Transformer-M [28]. As shown in Appendix Table 3, compared to the Transformer-based approach, our method increases the number of parameters by approximately 8%. Comparing to the EGNNs, although our number of parameters of the model is much larger than theirs, we can achieve faster training speed because some operators in EGNNs slow down training. Furthermore, to confirm that the improved model performance is not solely attributed to the larger model size, we experimented with a model with a similar size to EGNNs. Although the performance decreases compared to the larger model, it remains generally better and significantly faster than the other methods.

Table 3: Comparison of model size, training time, and performance between Geoformer and other models.

| Model | Size | Overall Training Time (GPU-hours) | MAE on $U_0$ | MAE on $U$ | MAE on $H$ | MAE on $G$ |
|---|---|---|---|---|---|---|
| SEGNN [4] | **1.03M** | 81 | 15 | 13 | 16 | 15 |
| TorchMD-NET [43] | 6.86M | 92 | 6.15 | 6.38 | 6.16 | 7.62 |
| Equiformer [24] | 3.53M | 61 | 6.59 | 6.74 | 6.63 | 7.63 |
| Transformer-M [28] | 47.4M | - | 9.37 | 9.41 | 9.39 | 9.63 |
| Geoformer | 50.1M | 55 | **4.43** | **4.41** | **4.39** | **6.13** |
| Geoformer-S | 6.4M | **20** | 5.20 | 5.12 | 5.19 | 6.78 |

# E    More Visualizations about Learned Interatomic Positional Encoding

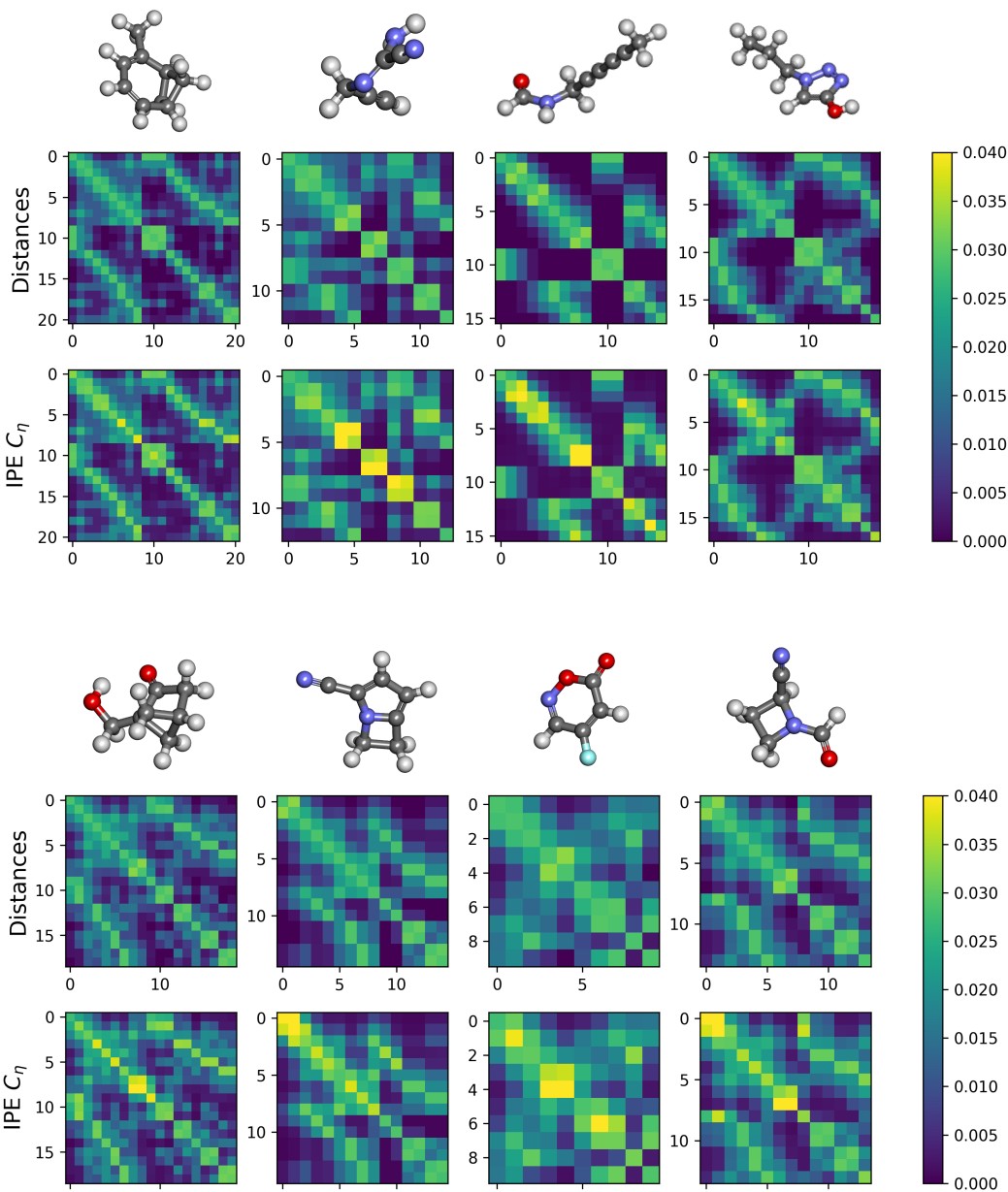

Figure 4: Visualization of IPE $C_\eta$ on molecules GDB75026, GDB9016, GDB52502, GDB126609, GDB106484, GDB24003, GDB23802 and GDB14823 in QM9 test set.

# F   Hyperparameters of Geoformer

Table 4: Hyperparameters for Geoformer trained on QM9 and Molecule3D

| Hyperparameters | QM9 | Molecule3D | |
|---|---|---|---|
| | | Random | Scaffold |
| Splits | Random | Random | Scaffold |
| Init learning rate | {1e-4, 1e-5, 2e-5} | 2e-4 | 2e-4 |
| Maximum epochs | 600 | 300 | 300 |
| LR warmup steps | 20000 | 10000 | 10000 |
| LR decay factors | 0.8 | 0.8 | 0.8 |
| LR patience | 15 | 5 | 5 |
| Early stopping patience | 150 | 30 | 30 |
| Weight decay | 0.0 | 0.0 | 0.0 |
| Batch size | 32 | 256 | 256 |
| RBF dimension | 64 | 32 | 32 |
| No. heads $H$ | 32 | 8 | 8 |
| No. layers $L$ | 12 | 9 | 9 |
| Embedding dimension $F$ | 512 | 256 | 256 |
| FFN embedding dimension $D$ | 2048 | 1024 | 1024 |

# G Details of Decoder

On QM9 and Molecule3D dataset, the property $\mu$ is calculated as follows:

$$\mu = \left\| \sum_{i=1}^{N} x_i^{out} (\vec{r}_i - \vec{r}_c) \right\| \tag{8}$$

where $\vec{r}_c$ denotes the center of mass, $x_i^{out}$ denotes the Encoder representation $x_i^L$ after passing through a two linear layers with SiLU activation function. Similarly, for the prediction of electronic spatial extent $\langle R^2 \rangle$, we use the following equation:

$$\langle R^2 \rangle = \sum_{i=1}^{N} x_i^{out} \|\vec{r}_i - \vec{r}_c\|^2 \tag{9}$$

For the remaining 10 properties $y$, we simply aggregate the final representation $x_i^{out}$ of atoms as follows:

$$y = \sum_{i=1}^{N} x_i^{out} \tag{10}$$