# OpenReview forum: "Geometric Transformer with Interatomic Positional Encoding"
_NeurIPS.cc/2023/Conference — NeurIPS 2023 poster_

### Official Review · Reviewer_mN7Q · 2023-06-15

**Soundness:** 3 good
**Presentation:** 3 good
**Contribution:** 2 fair
**Rating:** 5
**Confidence:** 4

**Summary:**

The paper proposes a Transformer based architecture incorporating interatomic information upon learned Atomic Cluster Expansion, integrated into the self-attention and refined residually over the network.
The method obtains state-of-the-art performance on a majority of prediction tasks on two popular molecular datasets.

**Strengths:**

The paper is well written.
The main novelty lies in the integration of learned ACE-based information into the model and its residual refinement.
The performances seem extremely significant.


**Weaknesses:**

I believe the most significant weakness of the paper is the lack of a literature review, which decreases the novelty level of the suggested method.

Transformer-based architectures for molecular property prediction is a field of research of at least three years old.
The manuscript cites only two related papers and compares its performance to a very low-accuracy Transformer model. \
To name a few of the firsts:\
Molecule attention transformer. arXiv:2002.08264, 2020\
Relative molecule self-attention transformer. arXiv:2110.05841, 2021\
3dtransformer: Molecular representation with transformer in 3d space. arXiv:2110.01191, 2021\
Geometry-aware transformer for molecular property prediction. arXiv:2106.15516, 2021\
Geometric transformer for end-to-end molecule properties prediction. arXiv:2110.13721, 2021

Thus, the proposed different embeddings and the integration of information into the self-attention are then not that novel.\
 For example, the initial IPE is similar to Physnet, and the integration of pairwise (also multi-scale) information into the transformer is similar to the references above (or other Transformer based propositions applied to other modalities).
Also, the proposed solution may be suboptimal compared to the literature and has not been ablated enough (e.g., learned or choice of handcrafted cut-off functions, the integration modality of the IPE into self-attention (beyond the addition presented in the Appendix's ablation); please see Questions).

Incorporating information from molecular mechanics/force field into neural networks has been proven as a powerful tool ([31,37,10,32 and many others]) by including beneficial inductive bias.
Thus, and as stated in the strengths, besides the very good results, I see the main contributions of the paper in the integration of augmented ACE information and also in the residual refinement of the IPE as the "neural" novelty.

Finally, the proposed model seems to have at least 20% higher capacity than the competing transformers and probably more compared to the others.

**Questions:**

1) From the ablation study, we can observe that variant 1 has the smallest impact on accuracy decrease among the two variants.
To better understand the impact of ACE/IPE, it would be nice to see the ablations of Variant 1 coupled with the residual procedure proposed in the paper (it should not be worse at least).

2) The Transformer literature (including molecular) provides many possible integrations of the IPE. One should provide more ablations rather than only with the standard sum (and besides the statements in line 194-)

3) Can you provide a capacity comparison with the other methods? The (significant) improvement in accuracy may also come from a higher model capacity and better and/or longer training. Also, regarding reproducibility, some details regarding the training are missing (e.g. number of epochs)

4) How do you explain the employment of a minimal number of expansions and harmonics still leads to such substantial results?
Also, in case such minimal hyperparameters are chosen, I believe one may simplify the cumbersome general case notation into a clear and easier-to-implement formulation.

5) The Weaknesses Section above should be addressed (e.g., novelty, comparisons).

Given the very good performance of the model, addressing these points may certainly rise the paper's impact and rating.

**Limitations:**

No specific/significant limitations are presented.

---

> ### Author Rebuttal · Authors · 2023-08-08
>
> ## Response
>
> Thanks for your valuable comments. We provide the point-to-point responses in the following.
>
> ### Weakness 1
>
> 1. Sorry for the lack of literature review torward the Transformer-based architecture. The works you mentioned are pioneers in the molecular modeling. We introduce them in the Introduction and Related Work, and obtain valuable insights from them to design some variants of positional encoding (see Question 5 for the novelty and comparison).
> 2. The key idea of different methods is try to design more effective positional encoding. According to your suggestion, we add these parts in the manuscript and conduct more ablation study (see Question 1 and 2).
> 3. Right. Thanks for your acknowledgement. In addition, the prevailing molecular mechanics/force field methods are based on EGNNs. Here we revisit the power of Transformers with the proposed IPE and we believe it is one of the novelty in this paper.
> 4. The introduction of IPE does introduce additional parameters. When we used the same settings as Transformer-M, such as the number of layers and hidden layer dimensions, we found that the parameters of the model went up a bit (about 8% higher capacity). Modern EGNN models are indeed relatively small (< 10M), however, due to the  CG-product or graph operations in their implementations, Transformer-based approach has a similar training speed to them (see Question 3 for details).
>
> ### Question 1
>
> Thanks for your comments. This suggested experiment does provide further evidence of the validity of our proposed framework for IPE. As you suggested, we update the initialized IPE using residuals that come from the transformation of the initial IPE itself, with no further information introduced, i.e. $C_{\eta}^{l+1} =  \operatorname{SiLU} \left(C_{\eta}^{l}W_{C}^l \right)  + C_{\eta}^{l}$
>
> The experimental results are as you expected, the residual update brings a slight performance gains (Non-updated IPE **0.563**; Residual IPE without additional information **0.551**; Geoformer **0.443**), while the main performance improvement comes from the additional information $\delta C_{\eta}^l = \sum_{\tau=1}^{\eta} W_{\tilde{B}}^l \sum_{v_{\tau}'} C_{vv_{\tau}'} \tilde{A_{vv_{\tau}'}}$  introduced to the IPE module.
>
> ### Question 2
>
> Thanks again for pointing these work. We add additional 3 variants (besides Question 1) brought from the ideas of above 4 papers:
>
> - for **Varaint 1** from *Molecule attention transformer*, we modify the IPE and attention block as:
>
> $$
> A=\left(\lambda_a \operatorname{SiLU}(\frac{QK^{T}}{\sqrt{d_k}})+\lambda_c\bf{C}_{\eta}\right)V
> $$
>
> setting $\lambda_a=\lambda_c=0.33$.
>
> - for **Variant 2** from *Geometry-aware transformer for molecular property prediction*, we modify the IPE following its paper:
>
> $$
> A=\frac{(QK^{T}\odot\bf{C}_{\eta})}{\sqrt{d_k}} \cdot V
> $$
>
> - for **Variant 3** from *Geometric transformer for end-to-end molecule properties prediction*, we modify the IPE following its paper:
>
> $$
> A=\operatorname{Softmax}(\frac{QK^{T}}{\sqrt{d_k}})\odot \bf{C}_{\eta}
> $$
>
> Due to the resource and time limitation, we only conduct the experiments on energy $U_0$:
>
> |  | Varaint 1 | Variant 2 | Variant 3 |
> | --- | --- | --- | --- |
> | U0 | 7.43 | 5.02 | 5.34 |
>
> We discovered that the variants involving the positional encoding **multiplicated** with *Key* and *Query* perform better than those using summation, implicitly supporting our theorem.
>
> ### Question 3
>
> We have provided the model size and overall training time in the table below, and to answer your question, we have also provided a model using the same size as EGNNs, and you can see that although the performance of the smaller model drops a bit compared to the larger model, the model is still SOTA when compared to the other methods. Following the previous studies, the maximum number of epochs is set to 600. We will add detailed settings in the appendix.
>
> |  | Model Size | Overall Training Time (GPU-hours) | MAE on U0 | MAE on U | MAE on H | MAE on G |
> | --- | --- | --- | --- | --- | --- | --- |
> | SEGNN | 1.03M | 81 | 15 | 13 | 16 | 15 |
> | TorchMD-NET | 6.86M | 92 | 6.15 | 6.38 | 6.16 | 7.62 |
> | Equiformer | 3.53M | 61 | 6.59 | 6.74 | 6.63 | 7.63 |
> | Transformer-M | 47.4M | - | 9.37 | 9.41 | 9.39 | 9.63 |
> | Geoformer | 50.1M | 55 | 4.43 | 4.41 | 4.39 | 6.13 |
> | Geoformer-S | 6.4M | 20 | 5.20 | 5.12 | 5.19 | 6.78 |
>
> ### Question 4
>
> In MACE [1], with $l=2$ could surpass most of the methods with higher order geometric tensors. Since the basis function matrix in Eq. 7 could represent the $\epsilon+1$ and $\epsilon+2$ body expansion simutaneously as well as the power of Transformers, we could use lower degree to achieve the comparable results. To obtain the rigor and universal derivation, we consider the higher degree of spherical harmonics.
>
> [1] Batatia, Ilyes, et al. "MACE: Higher order equivariant message passing neural networks for fast and accurate force fields." *Advances in Neural Information Processing Systems* 35 (2022): 11423-11436.
>
> ### Question 5
>
> All five studies endeavor to incorporate distance matrices for representing geometrical models, with certain investigations introducing the adjacency matrix (MAT) or chemical bond information (R-MAT) to further characterize geometries. Moreover, several studies examine the incorporation of attention bias; for instance, Molformer employs Adaptive PE to model molecules of varying sizes, GeoT forgoes softmax in favor of distance matrices as a scaling factor, and the application of multiplication for PE has been previously tried in the Geometric Transformer. Contrasting these  pioneer efforts, our methodology is grounded in the principles of ACE, providing a theoretical foundation for the empirical effectiveness of multiplication. Furthermore, as you mentioned, we introduced IPE based on ACE beyond pairwise distances, continuously refining it and bridging the performance gap between transformer-based approaches and EGNN.

---

> > ### Comment · Reviewer_mN7Q · 2023-08-10
> >
> > Thank you for your answers and new experiments, I think they really can improve the paper.
> >
> > I have a few remaining questions/comments:
> >
> > ### Weakness
> > 1 - Do you mean you "will" introduce them?
> >
> > ### Question 2
> > Thank you for running the experiments.
> > Variant 1 is similar to what you did in the ablations and has already been shown to underperform.
> > Regarding variant 2, there are some scaling and attention missing there.
> > Regarding variant 3, there should have been a learned parameterized mapping $\phi$ s.t. you multiply your attention by $\phi(C_{\mu})$ in order to learn the proper masking.
> >
> > ### Question 3
> > There are still some issues here. For example, PaiNN has 600K parameters and almost reaches Geoformer-S. Also, Transformer-M performance is very low and cannot really be taken into account in comparison while the performance of the other Transformers should be added too.
> >
> > To summarize, I still don't know if the very good performance comes from the high capacity of your model, the ACE or from $SiLU$/attention scaling. (I now know they are not due to the architecture residuals (question 1)). For instance, applying other Transformer models (e.g. the tested Variant 2-3 or others) with ACE may reach better performance at equal capacity.
> >
> > ### Question 4
> > I assume I can then summarize, as in the former review, that the main novelty comes from the integration of the ACE into existing Transformer architectures.

---

> > > ### Author Response · Authors · 2023-08-11
> > >
> > > Thank you for your active response! We provide the point-to-point responses for the remaining concerns in the following.
> > >
> > > ### Weakness 1
> > >
> > > We have introduced these works in our **Introduction**, **Related Work** and **Ablation section in the our revised manuscript**. The modified expression in our manuscript is as follows:
> > >
> > > > Several works have explored the integration of different types of positional encoding in Transformers. The MAT [1] and R-MAT [2] approaches methodologically introduce inter-atomic distances and chemical bond information as domain-specific inductive biases in Transformer architecture. Molformer [3] employs Adaptive PE to model molecules of varying sizes; GeoT [4] forgoes softmax in favor of distance matrices as a scaling factor, and the application of multiplication for PE has been previously tried in the Geometric Transformer [5].
> > >
> > > [1] Maziarka, Łukasz, et al. "Molecule attention transformer." *arXiv preprint arXiv:2002.08264* (2020).
> > >
> > > [2] Maziarka, Łukasz, et al. "Relative molecule self-attention transformer." *arXiv preprint arXiv:2110.05841* (2021).
> > >
> > > [3] Wu, Fang, Dragomir Radev, and Stan Z. Li. "Molformer: Motif-based transformer on 3d heterogeneous molecular graphs." *Proceedings of the AAAI Conference on Artificial Intelligence*. Vol. 37. No. 4. 2023.
> > >
> > > (Old name: 3dtransformer: Molecular representation with transformer in 3d space. arXiv:2110.01191, 2021)
> > >
> > > [4] Kwak, Bumju, et al. "Geometry-aware Transformer for molecular property prediction." *arXiv preprint arXiv:2106.15516* (2021).
> > >
> > > [5] Choukroun, Yoni, and Lior Wolf. "Geometric transformer for end-to-end molecule properties prediction." *arXiv preprint arXiv:2110.13721* (2021).
> > >
> > > ### Question 2
> > >
> > > Indeed, we have adapted our framework in strict accordance with the methods in the original paper. With respect to Variant 2, GeoT intentionally replaces $Softmax$ function with an alternative scaling method. In alignment with this, we use IPE to **replace** the aforementioned scaling function $\phi_w(D)$. With respect to Variant 3, since IPE is its residual updated and trainable, we have made a direct substitution of the original $\phi_w(D)$ with the IPE.
> > >
> > > In summary, our investigation encompasses a range of variants, comprising addition, weighted addition, no activation, multiplication after activation, as well as a variant presented in Question 1. Our choice, the "multiplication before activation" approach looks like the one that performance the best and is derived theoretically.
> > >
> > > ### Question 3
> > >
> > > 1. As per your suggestion, we have included a performance comparison with other Transformers as follows and we have added them as baseline in our revised manuscript. These comparisons highlight the effectiveness of our proposed method in the context of related Transformer-based approaches.
> > >
> > > |  | $\mu$ | $\alpha$ | HOMO | LUMO | gap | $R^2$ | ZPVE | U0 | U | H | G | Cv |
> > > | --- | --- | --- | --- | --- | --- | --- | --- | --- | --- | --- | --- | --- |
> > > | Molformer | 28 | 41 | 25 | 26 | 39 | 350 | 2.05 | 7.52 | 7.46 | 7.38 | 8.11 | 25 |
> > > | GeoT | 29.7 | 52.7 | 25.0 | 20.2 | 43.9 | 300.8 | 1.73 | 11.1 | 11.7 | 11.3 | 11.7 | 27.6 |
> > > | Geometric Transformer | 26.4 | 51 | 27.5 | 20.4 | 36.1 | 157 | 1.24 | 7.35 | 7.55 | 7.73 | 8.21 | 28.0 |
> > > | Geoformer | 10 | 40 | 18.4 | 15.4 | 33.8 | 27.5 | 1.28 | 4.43 | 4.41 | 4.39 | 6.13 | 22 |
> > >
> > > 2. Furthermore, as mentioned in Question 2, these Transformer-based approaches gained improvement of performance after introducing IPE (7.35 vs 5.34; 11.1 vs 5.02). In summary, the ablation studies among all variants demonstrate the effectiveness of the proposed IPE.
> > >
> > > 3. PaiNN's small number of parameters can be attributed to its simple operations (MLP). Moreover, EGNNs that utilize attentions block (see Equiformer and TorchMD-NET) tend to have a relatively large number of parameters as well. However, it is difficult to achieve further performance improvement by merely expanding PaiNN due to the over-smoothing problem. While the size of the model may contribute to the performance, we believe that the primary source of performance stems from the architectural updates and information provided by IPE. Moreover, we think the scaling between the model size and performance gains is a positive attribute. This advantage is particularly crucial in the rapidly evolving field of molecular modeling, where the ability to efficiently train large-scale models can lead to significant advancements in scientific discovery and practical applications.
> > >
> > > ### Question 4
> > >
> > > You're right. In this paper, our primary goal is to bridge the performance gap between Transformer-based methods, and EGNNs which include strong inductive bias, resulting in better performance. So, with the assistance of ACE theory, we **designed an architecture that achieves better performance** both **theoretically** and **practically** to **integrate the extra information IPE brings**. We appreciate your recognition of the performance and contributions in our paper.

---

> > > > ### Comment · Reviewer_mN7Q · 2023-08-16
> > > >
> > > > Thank you for your answers. While the "neural" contribution is not really important, I believe the good results, the new experiments and the updated related works improved the paper.

---

> > > > > ### Author Response · Authors · 2023-08-16
> > > > >
> > > > > Thanks for increasing your score. We've gone through the paper again, taking care of all your points, and it has shown more improvement. Our goal is to bring in more physics insights for Transformers. Introducing physics concepts into AI is bound to spark significant interest within the AI4Science community.

---

### Official Review · Reviewer_efGJ · 2023-06-23

**Soundness:** 3 good
**Presentation:** 3 good
**Contribution:** 2 fair
**Rating:** 5
**Confidence:** 3

**Summary:**

The authors propose an Interatomic Positional Encoding (IPE), and introduce the Geoformer. IPE is motivated by Atomic Cluster Expansion (ACE), which describes the many-body contributions in transformers for molecular modeling. The authors first derive integrated cluster potentials for atom pair i,j in a molecule, then leverage the interatomic potentials to propose the interactomic positional encoding. Further, the authors introduce the Geoformer architecture that implements IPE.

Experimentally, the authors compare Geoformer with popular molecular baselines on two datasets, QM9 and Molecule3D, and achieve the best results on most of the tasks. Additionally, the authors conduct ablation studies to varify the effectiveness of interatomic positional encoding.

Overall, the paper proposes a novel approach, and the experimental results show a better model performance than exisiting methods.

**Strengths:**

The approach inspired by Atomic Cluster Expansion is novel. Indeed, the positional encodings for modelling interatomic interactions have always been one of important factors in designing effective graph transformers for molecular data. Inspired by atomic cluster expansion, the authors propose a novel interatomic positional encoding that 'successfully' describes the interactions between atoms in a molecule, with theoretical justification. Experimentally, Geoformer surpass existing methods by a certain margin, which demonstrates the effectiveness of their approach. Overall, the paper presents a good quality of work with clear writing.

**Weaknesses:**

First, I find this paper is not so well-motivated. TorchMD-Net[1] model interatomic interactions by distance filters, and Muformer[2] leverage it and use additional 2D structures in the distance filters to model interatomic interactions. So, employing such information in Transformers is not completely underdeveloped. And it is unclear to me how ACE/IPE surpass previous methods in terms of effectiveness and efficiency.

In addition, the overall architecture suffers from numerical and training instabilities, for which the authors have also mentioned the limitation in section 5. Also, the authors do not compare the overall training time and model parameters in the experimental section, it is important to see the comparison of model sizes. Also, the error bars (standard deviations) are not shown in the tables. The authors do not perform sufficient experiments on proper datasets, it is also important to include experiments on MD17.

[1] Thölke, Philipp, and Gianni De Fabritiis. "Torchmd-net: equivariant transformers for neural network based molecular potentials." arXiv preprint arXiv:2202.02541 (2022).

[2] Hua, Chenqing, et al. "MUDiff: Unified Diffusion for Complete Molecule Generation." arXiv preprint arXiv:2304.14621 (2023).

**Questions:**

1. I would like to hear from authors about how ACE/IPE surpass the simplest positional encoding in [1] and [2] in terms of expressive power and calculation time. Moreover, how do the authors choose the trade-off between efficiency and effectiveness?

2. The authors mention that the architecture suffers from training instabilities. I wonder if there could be other techniques for stable training other than just using small learning rates.

3. I would like to see the comparison of model size (number of parameters) and training time, as well as the standard deviation over experimental runs.

4. I would like to see the comparison of model performance on the MD17 dataset, as MD17 provide more valid molecules. I also wonder why the authors do not perform experiments on MD17 in the beginning, as QM9 and MD17 are two of the most standard datasets for molecular simulation.


[1] Thölke, Philipp, and Gianni De Fabritiis. "Torchmd-net: equivariant transformers for neural network based molecular potentials." arXiv preprint arXiv:2202.02541 (2022).

[2] Hua, Chenqing, et al. "MUDiff: Unified Diffusion for Complete Molecule Generation." arXiv preprint arXiv:2304.14621 (2023).

---

> ### Author Rebuttal · Authors · 2023-08-08
>
> ## Response
>
> Thanks for your valuable comments. We provide the point-to-point responses at length in the following.
>
> ### Weakness 1
>
> 1. The distance filter (Radial Basis Fuctions, RBF) in TorchMD-Net is a sub-class of Atomic Cluster Expansion (ACE), which could be considered as the 2-body expansion. We mentioned it in Section 2.2, line 88. Here we add a detailed explanation. The basis function in ACE is a combination of radial basis functions (RBF) and spherical harmonics (SHs),  $\phi(r)=\sqrt{4\pi}R_{nl}(\Vert r \Vert)Y_l^m(r)$ in the original paper, with $n$ denoting the degree of RBF, $l$ denoting the degree of SHs, and $m$ denoting the index of SHs. If we set the $l=0$, the SH degrades to scalar $1$. Therefore, the basis functions remain as RBF, which is adopted by TorchMD-Net as the distance filter.
> 2. Compared to MUformer, which treated the 3D structure information as attention bias in attention block, we provide the theorical proof in Theorem 2, explaining the reason we directly multiply the proposed interatomic positional encoding (IPE) to the *Key* and *Query* from the perspective of the ACE. In a nutshell, we believe that employing the geometric information itself is not novel, but how to effectively utilize them is the most essential question. Concerns about instabilitiy, the comparison of model size and training as well as the standard deviations are replied in Question 2 and 3.
>
> ### Weakness 2
>
> Thank you for raising this concern. The main issue for us is that, for the MD17 experiment, it is required to use 0.7% of the data for training because they are generated from the same trajectory. This setup aims to test whether the machine learning force field is data-efficient. However, we think this is not suitable for Transformers, which prefer a large amount of data. Instead, we chose the prevailing QM9 and Molecule3D benchmarks, containing 130,831 and 3,899,647 molecules, respectively. However, as you suggested, we have conducted experiments on MD17 and display the energy and force error. The results show that, equipped with IPE, Transformer can also be applied to problems in limited-data scenarios.
>
> **Energy MAE**
>
> | Molecule | Geoformer | NequIP | TorchMD-Net | GemNet | PaiNN |
> | --- | --- | --- | --- | --- | --- |
> | Aspirin | **0.116** | 0.131 | 0.123 | - | 0.167 |
> | Ethanol | **0.051** | **0.051** | 0.052 | - | 0.064 |
> | Malonaldehyde | **0.074** | 0.076 | 0.077 | - | 0.091 |
> | Naphthalene | 0.087 | 0.113 | **0.085** | - | 0.116 |
> | Salicylic Acid | **0.093** | 0.106 | **0.093** | - | 0.116 |
> | Toluene | 0.078 | 0.092 | **0.074** | - | 0.095 |
> | Uracil | **0.095** | 0.104 | **0.095** | - | 0.106 |
>
> **Forces MAE**
>
> | Molecule | Geoformer | NequIP | TorchMD-Net | GemNet | PaiNN |
> | --- | --- | --- | --- | --- | --- |
> | Aspirin | **0.169** | 0.184 | 0.253 | 0.217 | 0.338 |
> | Ethanol | **0.063** | 0.071 | 0.109 | 0.085 | 0.224 |
> | Malonaldehyde | **0.115** | 0.129 | 0.169 | 0.155 | 0.319 |
> | Naphthalene | 0.043 | **0.039** | 0.061 | 0.051 | 0.077 |
> | Salicylic Acid | **0.088** | 0.090 | 0.129 | 0.125 | 0.195 |
> | Toluene | **0.044** | 0.046 | 0.067 | 0.060 | 0.094 |
> | Uracil | **0.066** | 0.076 | 0.095 | 0.097 | 0.139 |
>
> ### Question 1
>
> As discussed in Theorem 1, Remark, line 145, IPE with $v=1$ and  $\eta=1$ could be interpreted as introducing more geometric information like angle and dihedral with linear time complexity. In conjunction with the response to Weakness 1, we believe this explains why ACE/IPE outperforms those simple positional encoding, i.e., distance filters and the sum of Gaussian distances.
>
> ### Question 2
>
> The training instability is s common issue in EGNNs, like PaiNN and TorchMD-Net (e.g., see https://github.com/shehzaidi/pre-training-via-denoising/issues/3#issuecomment-1324959415) as well as other Transformers. However, to ensure the **equivariance** of the basis functions, we **cannot** directly apply *LayerNorm* to them. Here attempt to linearly shrink the norm of the basis over the last dimension to stablize training.
>
> ### Question 3
>
> We list the standard deviations when we repeated an additional 2 trials on the QM9 dataset, and from the results it can be seen that the model is not particularly sensitive to the partition of the dataset and the initial parameters.
>
> |  | $\mu$ | $\alpha$ | HOMO | LUMO | gap | $R^2$ | ZPVE | U0 | U | H | G | Cv |
> | --- | --- | --- | --- | --- | --- | --- | --- | --- | --- | --- | --- | --- |
> | Std | 0.38 | 1.65 | 0.51 | 0.73 | 0.76 | 0.69 | 0.05 | 0.15 | 0.14 | 0.13 | 0.13 | 0.41 |
>
> We directly used the model sizes and training times reported in Equiformer and Transformer-M, and experimented with Geoformer on a NVIDIA V100 GPU. It can be seen that in contrast to the Transformer-based method, our method increases the number of parameters by about 8%(47.4M vs 50.1M). Comparing to the EGNNs, although our number of parameters of the model is much larger than theirs, we can achieve faster training speed because some operators in EGNNs slow down training. In addition, to verify that the improvement in model performance does not only come from the larger model size, we also experimented with the model using a similar size as EGNNs, and although the performance drops compared to the larger model, it is still generally better and significantly faster than the other methods.
>
> |  | Model Size | Overall Training Time (GPU-hours) | MAE on U0 | MAE on U | MAE on H | MAE on G |
> | --- | --- | --- | --- | --- | --- | --- |
> | SEGNN | 1.03M | 81 | 15 | 13 | 16 | 15 |
> | TorchMD-NET | 6.86M | 92 | 6.15 | 6.38 | 6.16 | 7.62 |
> | Equiformer | 3.53M | 61 | 6.59 | 6.74 | 6.63 | 7.63 |
> | Transformer-M | 47.4M | - | 9.37 | 9.41 | 9.39 | 9.63 |
> | Geoformer | 50.1M | 55 | 4.43 | 4.41 | 4.39 | 6.13 |
> | Geoformer-S | 6.4M | 20 | 5.20 | 5.12 | 5.19 | 6.78 |
>
> ### Qustion 4
>
> Same as Weakness 2.

---

> > ### Comment · Reviewer_efGJ · 2023-08-14
> > **Response by Reviewer**
> >
> > The authors have addressed my concerns in a respectful way. I will change my score from 4 to 5. I hope the authors can properly cite [1][2] in their revised version.
> >
> > [1] Thölke, Philipp, and Gianni De Fabritiis. "Torchmd-net: equivariant transformers for neural network based molecular potentials." arXiv preprint arXiv:2202.02541 (2022).
> >
> > [2] Hua, Chenqing, et al. "MUDiff: Unified Diffusion for Complete Molecule Generation." arXiv preprint arXiv:2304.14621 (2023).

---

> > > ### Author Response · Authors · 2023-08-14
> > >
> > > We appreciate your constructive feedback and are glad to have addressed your concerns effectively. As per your suggestion, we have introduced and cited the following two papers in our revised manuscript:
> > >
> > > > Several works have incorporated interatomic interactions into molecular modeling. For instance, TorchMD-Net [1] included the radial basis functions (RBF) as distance filter to the attention matrix. MUformer [2] further extended the distance filter by incorporating additional 2D structural information, resulting in improved performance and applicability to molecule 2D-3D co-generation.
> > > >
> > >
> > > [1] Thölke, Philipp, and Gianni De Fabritiis. "Torchmd-net: equivariant transformers for neural network based molecular potentials." arXiv preprint arXiv:2202.02541 (2022).
> > >
> > > [2] Hua, Chenqing, et al. "MUDiff: Unified Diffusion for Complete Molecule Generation." arXiv preprint arXiv:2304.14621 (2023).
> > >
> > > We sincerely hope that our revised manuscript will make a meaningful contribution to the research community. Thank you for your support and recognition of our efforts.

---

### Official Review · Reviewer_z5Cf · 2023-07-03

**Soundness:** 3 good
**Presentation:** 3 good
**Contribution:** 3 good
**Rating:** 6
**Confidence:** 2

**Summary:**

The paper introduces Geoformer, a new geometric transformer for molecular property prediction. The authors argue that while transformers have been dominant in various data modalities, their application to molecular modeling has been limited. To address this, the authors propose Interatomic Positional Encoding (IPE) based on atomic cluster expansion (ACE) theory, which captures complex interactions within molecules. They evaluate Geoformer on the QM9 dataset and the Molecule3D dataset, demonstrating its superior performance compared to the existing methods.

**Strengths:**

I am unfamiliar with this topic, but the idea of capturing complex geometric features by introducing Interatomic Positional Encoding seems novel. The motivation is well explained, and the authors also provide some mathematical proofs. The presented positional encoding may contribute to this field.

The results on the QM9 dataset and the Molecule3D dataset show that Geoformer outperforms other competitors in terms of most metrics. This highlights the effectiveness of the proposed method in capturing and utilizing valuable geometric information.


**Weaknesses:**

I did not see a major weakness, but I have some minor concerns about the experiments.

Although this paper handles the problem of molecular property prediction, it seems that the presented geometric transformer block could potentially serve as a plugin in other machine learning and computer vision tasks. For example, the task of point cloud analysis could also benefit from describing complex geometric features. It would be beneficial and convincing if the authors could show the effectiveness of the presented method in such a context.

The presented positional encoding includes some learnable parameters, which is different from the traditional encoding strategy, and the encoding is dynamically updated. Therefore, the ablation study towards the effectiveness of the presented method is crucial. However, the authors put some results in the appendix instead of the main paper, which downgrades the importance. It would be better if the authors could shed more light on the ablation study in the main paper.


**Questions:**

Please refer to the weaknesses for my concerns about the potential applications of this method.

**Limitations:**

The method could be computationally expensive as the presented positional encoding involves additional learnable parameters.

---

> ### Author Rebuttal · Authors · 2023-08-08
>
> ## Reponse
>
> Thanks for raising some valuable concerns. We provide the point-to-point response in the following.
>
> ### Weakness 1
>
> After conducting a literature review on point cloud analysis, we discovered related work which incorporates geometric information in the point cloud modeling [1-5]. Though few of these studies consider *equivariance* as in molecules, there are several works like TFN [7] are famous, and the most relavent work would be by Qin et al. [6], who propose *Pair-wise Distance Embedding* and *Triplet-wise Angular Embedding* in Transformers as attention bias to *Key*. Therefore, we believe our proposed interatomic positional encoding (IPE) have the potential to generalize to the field of point cloud analysis beyond the molecule.
>
> Due to the time limitation, we cannot present more results in such a context, but would investigate its effectiveness in the future. Thanks for your valuable insights again!
>
> [1] Yu, Xumin, et al. "Pointr: Diverse point cloud completion with geometry-aware transformers." *Proceedings of the IEEE/CVF international conference on computer vision*. 2021.
>
> [2] Ma, Xu, et al. "Rethinking network design and local geometry in point cloud: A simple residual MLP framework." *arXiv preprint arXiv:2202.07123* (2022).
>
> [3] Chen, Zhi, et al. "Sc2-pcr: A second order spatial compatibility for efficient and robust point cloud registration." *Proceedings of the IEEE/CVF Conference on Computer Vision and Pattern Recognition*. 2022.
>
> [4] Yu, Hao, et al. "Rotation-invariant transformer for point cloud matching." *Proceedings of the IEEE/CVF Conference on Computer Vision and Pattern Recognition*. 2023.
>
> [5] Hou, Ji, et al. "Mask3D: Pre-training 2D Vision Transformers by Learning Masked 3D Priors." *Proceedings of the IEEE/CVF Conference on Computer Vision and Pattern Recognition*. 2023.
>
> [6] Qin, Zheng, et al. "Geometric transformer for fast and robust point cloud registration." *Proceedings of the IEEE/CVF conference on computer vision and pattern recognition*. 2022.
>
> [7] Thomas, Nathaniel, et al. "Tensor field networks: Rotation-and translation-equivariant neural networks for 3d point clouds." arXiv preprint arXiv:1802.08219 (2018).
>
> ### Weakness 2
>
> Due to the page limitation, we had to put some ablation experiments in the Apendix. However, we highlight the analysis of ablation study in the manuscript as you suggested.
>
> ### Limitation 1
>
> Indeed, our approach introduces additional computational overhead. Compared with Transformer-M using the same setting (the number of layers and hidden dimensions), our model increases by less than 8% parameters.  However, considering the improvement in performance, we believe the extra computational overhead is acceptable.

---

> > ### Comment · Reviewer_z5Cf · 2023-08-14
> >
> > I would like to thank the authors for the point-to-point response. I appreciate the effort the authors have done towards the potential applications. I believe this paper would be more solid if those comparisons could be incorporated. I would like to keep my original score based on the rebuttal.

---

> > > ### Author Response · Authors · 2023-08-14
> > >
> > > We are unfamiliar with point cloud datasets, which makes it challenging for us to finish training and report results during the rebuttal process. However, this is a highly promising direction.
> > >
> > > Regarding other points, we have conducted additional ablation experiments and emphasized them in our revised manuscript. We have also included a comparison of model sizes. Furthermore, we have conducted experiments on another molecular dataset, MD17. For further details, please refer to our **Response to Reviewer mN7Q** and **Response to Reviewer efGJ**. These results have made our paper more solid.
> > >
> > > Thank you for your comments, which have strengthened our article.

---

### Official Review · Reviewer_nHVJ · 2023-07-05

**Soundness:** 4 excellent
**Presentation:** 3 good
**Contribution:** 4 excellent
**Rating:** 8
**Confidence:** 5

**Summary:**

The paper introduced a Transformer-based model with interatomic positional encoding (IPE) for molecular modeling, which the authors termed "Geoformer". Geoformer incorporated novel positional encoding derived from empirical physical knowledge (atomic cluster expansion) and was able to achieve comparable or even superior performance on multiple benchmarks.

**Strengths:**

1. To the best of my knowledge, the proposed approach for injecting physical knowledge, i.e., the atomic cluster expansion into the positional encoding in Transformers is novel in this field, which is dominated by EGNNs. The extensive experiments and ablation studies also demonstrated the expressiveness of the proposed architecture.

2. The idea of merging clusters further considers the interaction/potential between different clusters, while the original ACE only considers one individual cluster.

3. The proposed method has advantages over current EGNNs such as incorporating angles and dihedrals, and seems more computationally efficient because the calculation between clusters is still under linear time complexity.

4. It revisits the Transformer architecture in the field of molecular modeling and outperforms most of the current sota models in QM9 and Molecule3D benchmark.

**Weaknesses:**

1. Some notations and figures about the merged cluster are a bit confusing and can be improved (see the following question section).

2. The authors claimed the proposed model was computationally more efficient and memory-saving. The authors can provide quantitative data for a clearer justification (see the following question section).

**Questions:**

### Notations about the merged cluster.

1. The relation between Fig.1 and Eq. 7 is confusing. In Eq. 7, the author shows the matrix of $A$-basis to represent the new basis for the merged cluster, while in Fig. 1, it turns out to be the matrix of $B$-basis to describe the proposed positional encoding. Their relation needs to be further explained in the manuscript.
2. It is hard to follow the relation between the basis of the merged cluster with the proposed interatomic positional encoding for those readers who are not familiar with ACE. Also, only after reading several times could I realize that the matrix of $A$-basis in Theorem 1 is the additional (right) term in Eq.7. The relation between Theorem 1 and 2 should be clarified.
3. What is the difference between $A_{i(i)}$ and $A_{i}$?
4. Following Q3, what is the difference between $A_{i(i)}$ in the merged cluster with the original $A$-basis in (linear) ACE?

### Complexity

5. The author mentioned they only utilized reduced settings, i.e., body-expansion=1 and order of spherical harmonics=1. What is the time/memory complexity for the algorithm?
6. Compared with the explicit calculation like GEM [1], and those approaches like SphereNet [2] which employ the CG-product, how much time/memory would Geoformer reduce?
7. What is the model size of Geoformer compared with other EGNNs?

### Other minor problems

8. The index of interatomic positional encoding (C) in the left corner, Fig. 1 is wrong.
9. how is the molecule graph being padded?



[1] Liu, Lihang, et al. "GEM-2: Next Generation Molecular Property Prediction Network by Modeling Full-range Many-body Interactions." (2022).

[2] Liu, Yi, et al. "Spherical message passing for 3d graph networks." *arXiv preprint arXiv:2102.05013* (2021).


**Limitations:**

The authors have adequately addressed the limitation and potential negative societal impact in Section 5.

---

> ### Author Rebuttal · Authors · 2023-08-08
>
> ## Response
>
> Thanks for your comments and acknowledgment of our work. We provide the point-to-point responses in the following.
>
> ### Question 1
>
> We appreciate your suggestion and have provided more explanation of the construction of the B-basis in Theorem 2 and the Appendix for better clarity. Indeed, the merged B-basis remains in matrix form, similar to the merged A-basis matrix. It is additionally multiplied with CG-coefficients to ensure permutation and isometry-invariance.
>
> ### Question 2
>
> Thank you for pointing this out. We understand that readers unfamiliar with ACE or the physics of harmonics might find it challenging to follow. We have added more explanation of the connection between Theorem 1 and Theorem 2. The central point is that "the potential of the merged cluster in Theorem 1 is the physical explanation of the interatomic positional encoding in Theorem 2." In Section 3.1, we demonstrate how to incorporate IPE within the Transformer in practice. We aim to introduce the proposed IPE both theoretically and practically.
>
> ### Question 3 and 4
>
> $A_i$ is the original A-basis in ACE. Since we conduct the cluster merging, $A_{i(i)}$ denotes the multiplication of two $A_i$, describing the $\epsilon+1$ body expansion in the manscript line 137. Together, the matrix in Eq. 7 could represent the $\epsilon+1$ and $\epsilon+2$ body expansion simutaneously.
>
> ### Question 5 and 6
>
> Due to the linear ACE theory, the complexity of constructing of interatomic positional encoding is $\mathcal{O}(N)$ with $N$ denoting the number of atom with one molecule. In contrast, the explicit extraction method such as GEM, GemNet and SphereNet have a complexity of $\mathcal{O}(N^{3})$ due to extracting dihedral angles for 4-body interaction. The main computational consumption lies in the Attention block, with a complexity of $\mathcal{O}(N^{2})$. We cannot replace this part with the prevailing linear attention at this moment because we have to update learnable IPE. We would explore these possibilities in the future work. Compare to higher-order of spherical harmonics, which require the pre-computed CG-coefficients and products, we streamline this part by reduced setting. Since the number of spherical harmonics is $(l+1)^2$ with degree $l$, it would reduce consumption (tensor paths) by approximately $Nl$.
>
> ### Question 7
>
> Modern EGNN models are indeed relatively small (50.7M vs 10M), however, due to the  CG-product or graph operations in their implementations, Transformer-based approach has a similar training speed to them (see the below table for details).
>
> |  | Model Size | Overall Training Time (GPU-hours) | MAE on U0 | MAE on U | MAE on H | MAE on G |
> | --- | --- | --- | --- | --- | --- | --- |
> | SEGNN | 1.03M | 81 | 15 | 13 | 16 | 15 |
> | TorchMD-NET | 6.86M | 92 | 6.15 | 6.38 | 6.16 | 7.62 |
> | Equiformer | 3.53M | 61 | 6.59 | 6.74 | 6.63 | 7.63 |
> | Transformer-M | 47.4M | - | 9.37 | 9.41 | 9.39 | 9.63 |
> | Geoformer | 50.1M | 55 | 4.43 | 4.41 | 4.39 | 6.13 |
> | Geoformer-S | 6.4M | 20 | 5.20 | 5.12 | 5.19 | 6.78 |
>
> ### Minor Problem 1
>
> Thanks for reaching this out. We have modified the $C_{\gamma}$ with $C_{\eta}$ in Fig. 1.
>
> ### Minor Problem 2
>
> In one batch, we select the the maximum number of molecular atom, and pad other molecules with $0$, detaching the gradient at these padding indices in *Embedding* layer and add *key_padding_mask* for computing attention correctly.

---

> > ### Comment · Reviewer_nHVJ · 2023-08-10
> > **Comment on Authors' Rebuttal**
> >
> > I appreciate your comprehensive response regarding my questions and concerns. Based on the rebuttal, I believe you have appropriately and adequately addressed all of my concerns on the notation, formulation, and complexity of the model. The theoretical analysis and the additional experiments regarding model sizes and computational resources further justified the efficiency of your proposed model, and I hope you will add this result to the revised manuscript to make it more concrete.
> >
> > In conclusion, I believe that, after properly addressing the above questions, this work provided a novel architecture of a Transformer-based equivariant GNN, with relatively extensive experiments and ablation studies to demonstrate the superior performance over the baselines and computational efficiency compared to normal EGNNs. In this regard, I am happy to raise my original score from 7 to 8.

---

> > > ### Author Response · Authors · 2023-08-11
> > >
> > > Thank you for recognizing our paper and raising your score. We have incorporated these changes to enhance our paper in our revised manuscript.

---

### Official Review · Reviewer_M4bW · 2023-07-06

**Soundness:** 3 good
**Presentation:** 2 fair
**Contribution:** 2 fair
**Rating:** 5
**Confidence:** 3

**Summary:**

The paper proposes a geometric Transformer called Geoformer for geometric molecular modeling. It designs Interatomic Positional Encoding (IPE), taking 3D geometric information into account to parametrize atomic environment for positional encoding in Transformer architecture. Geoformer adopts a learned Interatomic Positional Encoding (IPE) to the attention block of the Transformer. Geoformer is evaluated on QM9 and Molcule3D datasets with various tasks and shows superior performance compared with Transformers and EGNN baselines.

**Strengths:**

- Geoformer effectively integrates domain knowledge from physics, specifically Atomic Cluster Expansion (ACE), into the machine learning field for molecular property prediction. The Interatomic Positional Encoding (IPE) with ACE offers a robust method for incorporating geometric priors from other scientific disciplines into the positional encoding of Transformers.
- IPE reflects geometric information and interatomic relations within the Transformer architecture. Leveraging IPE, Geoformer demonstrates strong performance on 3D molecular property prediction tasks across QM9 and Molecule3D datasets over various baselines.


**Weaknesses:**

Overall, the representation of this paper could be polished further to improve accessibility. Considering this paper is targeted at a machine learning conference, more detailed explanations about the background (ACE) would be beneficial for potential readers from this community. If the space matters, the appendix would be a good place for the additional information.
- For example, the term "cluster" is used in Theorem 1 without a clear definition. I presume this is the concept introduced in ACE as well.
- Although there are some introductory sentences, Section 3.1 only consists of two theorems without any proper explanation of why these theorems are needed and how they can be used.  For example, Theorem 1 is introduced without any prior explanation of its necessity. It would be better to explain why Theorem 1 is needed and its main messages beforehand.
- The attention block introduced through Equations 18 to 23 seems like an important contribution of this paper. However, with the given content, it is unclear what intuition leads to the specific form of learnable IPE in Eq 20, 21.
- Preliminary and related work sections need to be separated for better explanation. Currently, section 2.1 and 2.3 is more like related work. I suggest moving these to the appendix and providing more information on ACE in the preliminary.
- It is questionable whether Theorem 2 is indeed a theorem; it seems like a matrix IPE matrix C was defined for positional encoding. More explanation would be appreciated.


**Questions:**

- In line 163, is A_{i, v_{tau}} defined in the same way as equation 3? If so, the bold font should be unified.
- Is C_{vv^{prime}_{tau}} the same as C_{vv^{prime}} in the equation 4?
- In subsection 4.5 and figure 3, is there any task-related or domain-specific ground truth where IPE focuses more on the atomic relationship in learned IPE? It is necessary for the claim that "IPE further enhances signals and shows significant distinction" to be persuasive.


**Limitations:**

The limitations are well addressed in the paper.

---

> ### Author Rebuttal · Authors · 2023-08-07
>
> ## Response
>
> Thanks for your valuable comments. We provide point-to-point responses in the following:
>
> ### Weakness 1
>
> The term **cluster** is a concept in the original Atomic Cluster Expansion (ACE) theory, which represents the local chemical environment of centered atoms. A cluster $\alpha$ contains one centered atom $i$, $K$ neighbor atoms (elements, denoted as $j$) with $K$ bonds. We’ve added it in the Appendix.
>
> ### Weakness 2
>
> We apologize for the confusion. Theorem 1 serves as the foundation for Theorem 2 and the architecture of Geoformer in Section 3. It describes the proposed interatomic positional encoding (IPE) from a physics perspective. Specifically, we demonstrate how to construct the IPE (Theorem 2) **in theory**, representing the potential of merged cluster (Theorem 1). Next, we illustrate how to leverage the IPE (Theorem 2) within traditional Transformers (Section 3.1) **in practical**. In summary, we aim to demonstrate the proposed IPE both theoretically and practically.
>
> ### Weakness 3
>
> We apologize for the oversight. The concepts behind Eq. 20 and 21 originate from AlphaFold2, which utilizes pairwise embedding $z$ with an activation function as a learnable gate, combined with the previous residual. We recommend referring to the *row/column-wise gated self-attention* in AlphaFold2 supplementary. We have also added this reference to the manuscript.
>
> ### Weakness 4
>
> Following your suggestion, we separate the sections into **Preliminaries** and **Related Work** and provide a more detailed introduction of ACE.
>
> ### Weakness 5
>
> In Theorem 2, we aim to prove why the interatomic positional encoding should be multiplied with the Key and Query in the traditional Transformers, as opposed to other operations such as attention bias or distance filter. Therefore, we consider this construction and proof as a theorem. Additionally, we have included the complete proof in the Appendix.
>
> ### Question 1
>
> Thanks for reaching this out. We have removed the bold font $i$ in line 163.
>
> ### Question 2
>
> Right. They both represent the Clebsch-Gordan coefficients to ensure the permutation and isometry-invariance.
>
> ### Question 3
>
> We appreciate your comments. To testify “IPE further enhances signals and shows significant distinction”, in Figure 3, we compare the IPE with the distance signal, which the most of previous work adopted in the original manuscript to show how IPE functions. Based on your suggestion, we are exploring for some domain-specific ground truth, e.g., electron density, to further substantiate the effectiveness of the proposed IPE. More results would be added if we find something interesting.

---

> > ### Comment · Reviewer_M4bW · 2023-08-17
> >
> > After reading the other reviews and responses, I conclude that this work is worth being introduced to a wider audience at the conference. As mentioned earlier, most of my concerns are about the presentation and clarity of the manuscript. Although it is not allowed to modify the manuscript during rebuttal, I hope the authors will well address these concerns and improve their manuscript for the wider audience. I'll adjust my score from 4 to 5.

---

> > > ### Author Response · Authors · 2023-08-17
> > >
> > > We greatly appreciate your decision to raise the score for our paper. We have meticulously revised the manuscript in accordance with your comments, ensuring that the revised version is accessible to the wider machine learning community. Once again, we are grateful for your invaluable feedback, which has significantly contributed to enhancing the quality of our work.

---

### Author Response · Authors · 2023-08-17
**Summary of the Rebuttal Process**

Dear ACs, PCs and all reviewers,


We would like to express our gratitude to all the reviewers for their valuable comments and feedback on our work. In this summary, we aim to provide the ACs/PCs with a clear understanding of the changes made during the rebuttal process:

All reviewers recognized the novelty and quality of our manuscript.

- Per Reviewer **M4bW**'s suggestion, we have added more background information on ACE to make it more accessible to a wider audience.
- We have addressed questions regarding the theory and details of our interatomic positional encoding (IPE) raised by Reviewers **nHVJ**, **efGJ**, and **mN7Q**. All concerns have been satisfactorily resolved.
- We have done more literature review on point cloud analysis and Transformers suggested by Reviewer **z5Cf** and **mN7Q.**
- We have conducted additional experiments on the MD17 benchmark and explained why it was not initially included, as requested by Reviewer **efGJ**. Furthermore, we have added a model size comparison (suggested by Reviewers **nHVJ**, **efGJ**, and **mN7Q**), standard deviations in QM9 (Reviewer **efGJ**), various ablations (Reviewer **mN7Q**), and more comparisons with some Transformer baseline (Reviewer **mN7Q**). In total, we supplemented about **40** model training experiments. All concerns have been adequately addressed.

Following the rebuttal with reviewers, we have successfully resolved all concerns in a respectful manner. We appreciate the reviewers' contributions to improving the quality of our manuscript.


Best,

Authors

---

### Decision · Program_Chairs · 2023-09-21

**Decision:**

Accept (poster)

**Comment:**

The paper tries to improve application of popular transformer architecture for molecular property prediction. In this regards, the authors incorporate domain knowledge from physics, specifically Atomic Cluster Expansion (ACE), into the transformer attention mechanism and positional embeddings. Empirically, it bridges the performance gap between EGNN and Transformer. Reviewers found the paper to be technically solid and demonstrates strong results, but also raised concerns regarding novelty, model complexity, and presentation issues like notation, background etc.

We thank the authors and reviewers to actively engage during the discussion phase to improve the paper including providing more ablation experiments, which were also intriguing.

- The "neural" contribution/novelty is limited and the main contribution seems to be the integration of the ACE into existing (very overparameterized) Transformer architecture, similar to what was previously done with other pairwise metrics or handcrafted descriptors but unlike those, ACE seems to have such a large impact on the accuracy. Thus contribution of the paper should be updated.

- In terms of writing, several reviewers pointed out issues with background knowledge, and missing related work. After the rebuttal, the authors provided revised paragraphs in response to them. Please incorporate these including comparison with other transformers and continue to refine the manuscript.